# Accelerating Langevin Monte Carlo via Efficient Stochastic Runge–Kutta Methods beyond Log-Concavity

**Bin Yang** [1]  **Xiaojie Wang** [1]

## Abstract

Sampling from a high-dimensional probability distribution is a fundamental algorithmic task arising in wide-ranging applications across multiple disciplines, including scientific computing, computational statistics and machine learning. Langevin Monte Carlo (LMC) algorithms are among the most widely used sampling methods in high-dimensional settings. This paper introduces a novel higher-order and Hessian-free LMC sampling algorithm based on an efficient stochastic Runge–Kutta method of strong order $1.5$ for the overdamped Langevin dynamics. In contrast to the existing Runge–Kutta type LMC of Li et al. (2019) involved with three gradient evaluations, the newly proposed algorithm is computationally cheaper and requires only two gradient evaluations at each iteration. Under certain log-smooth conditions, non-asymptotic error bounds of the proposed algorithms are analyzed in $\mathcal{W}_2$-distance. In particular, a uniform-in-time convergence rate of order $O(d^{\frac{3}{2}} h^{\frac{3}{2}})$ is derived in a non-log-concave setting, matching the convergence rate proved in the aforementioned work under the log-concavity condition. Numerical experiments are finally presented to demonstrate the effectiveness of the new sampling algorithm.

## 1. Introduction

Recent years have witnessed substantial progress in the sampling problem from a given high-dimensional probability distribution of the form $\pi(\mathrm{d}x) \propto e^{-U(x)}\mathrm{d}x$, where $U \colon \mathbb{R}^d \to \mathbb{R}$ is a potential function. Such a problem finds many applications in diverse fields such as Bayesian statistics (Cotter et al., 2013), machine learning (Andrieu et al.,

[1] School of Mathematics and Statistics, HNP-LAMA, Central South University, Changsha, China . Correspondence to: Xiaojie Wang <x.j.wang7@csu.edu.cn, x.j.wang7@gmail.com>.

*Proceedings of the 43rd International Conference on Machine Learning*, Seoul, South Korea. PMLR 306, 2026. Copyright 2026 by the author(s).

2003) and molecular dynamics (Lelievre & Stoltz, 2016). Langevin Monte Carlo (LMC) sampling methods are among the most widely used algorithms in high-dimensional sampling. A typical unadjusted LMC is given by

$$\hat{Y}_{n+1} = \hat{Y}_n - \nabla U(\hat{Y}_n)h + \sqrt{2h}\zeta_{n+1}, \quad \hat{Y}_0 = \xi, \quad (1)$$

where $\zeta_n := (\zeta_n^1, \zeta_n^2, \cdots, \zeta_n^d)^T$, $n \in \mathbb{N}$, are i.i.d. standard $d$-dimensional Gaussian random variables. This algorithm can be regarded as the Euler–Maruyama method for the following (overdamped) Langevin stochastic differential equation (SDE):

$$\mathrm{d}X_t = -\nabla U(X_t)\,\mathrm{d}t + \sqrt{2}\,\mathrm{d}W_t, \quad X_0 = \xi,\, t > 0, \quad (2)$$

where $W. := (W_.^1, W_.^2, \cdots, W_.^d)^T \colon [0, \infty) \times \Omega \to \mathbb{R}^d$ is $d$-dimensional Brownian motion defined on the filtered probability space $(\Omega, \mathcal{F}, \{\mathcal{F}_t\}_{t \geq 0}, \mathbb{P})$, satisfying the usual conditions. The initial data $\xi \colon \Omega \to \mathbb{R}^d$ is assumed to be $\mathcal{F}_0$-measurable. Under mild conditions, the Langevin SDE admits $\pi$ as its unique invariant distribution (see, e.g., Pavliotis (2014)). Therefore, one can expect the samples generated by (1) approximately follow the target distribution $\pi$, after long-time iterations. Over the past decade, there have been a number of works devoted to analyzing non-asymptotic error bounds in various distances of the classical LMC algorithm (1), under log-concavity and non-log-concavity conditions. Under a strongly log-concave condition ($m > 0$):

$$\langle x - y, \nabla U(x) - \nabla U(y) \rangle \geq m|x - y|^2, \ \forall x, y \in \mathbb{R}^d \ (3)$$

and the gradient Lipschitz condition (22), the LMC (1) admits non-asymptotic convergence of order $O(\sqrt{dh})$ in both total variation and $\mathcal{W}_2$-distance (Durmus & Moulines, 2017; Dalalyan, 2017b;a; Durmus et al., 2019; Cheng & Bartlett, 2018). The order one-half can be promoted to be $O(dh)$, by additionally imposing the Hessian Lipschitz condition (24) on $U$ (Durmus & Moulines, 2019). Under a kind of linear growth condition of the 3rd-order derivative, the linear dimension dependence can be further improved to be $O(\sqrt{d}h)$ (Li et al., 2022; Altschuler & Chewi, 2024a).

However, the log-concave condition (3) is restrictive and hardly satisfied in practice. Recently, much progress has been made in deriving non-asymptotic error bounds of the

classical LMC (1) in non-log-concave settings such as the convexity at infinity condition and the log-Sobolev inequality (Cheng et al., 2018; Majka et al., 2020; Mou et al., 2022; Altschuler & Chewi, 2024a; Yang & Wang, 2025; Li et al., 2025a; Pang et al., 2025; Chewi et al., 2024; Li et al., 2025b; Pagès & Panloup, 2023; Mousavi-Hosseini et al., 2023; Altschuler & Chewi, 2024b), see, e.g., Yang and Wang (2025) for a literature review. Without the log-concave condition (3), the long-term error analysis is much more challenging and a lot of efforts have been spent to obtain the desired non-asymptotic error bounds of order $O(\sqrt{d}h)$ under the gradient Lipschitz condition and order $O(dh)$ or $O(\sqrt{d}h)$ under smoother conditions.

From the above discussion, one can readily observe that the best convergence rate the algorithm (1) attains is order one, even under very smooth conditions. This is expected as the scheme (1) is nothing but an Euler method for (2). A natural and interesting question thus emerges:

**(Q1).** *Are there any higher-order and easily implementable schemes to accelerate the Langevin Monte Carlo sampling?*

A naive attempt is to use the strong Taylor scheme (7), which stems from a truncation of Itô Taylor expansions of the solution to the Langevin SDE (2), up to strong order 1.5. It was proved in Sabanis and Zhang (2019) that the Taylor scheme (7) admits a non-asymptotic error bound of order $O(d^2h^{3/2})$ in $\mathcal{W}_2$-distance under a strong convexity assumption. More recently, Neufeld and Zhang (2024) established a non-asymptotic error estimate for the Taylor scheme in $\mathcal{W}_1$-distance of order $O(e^{O(d)}h^{3/2})$ in a non-convex setting, as a consequence of which, they also obtained an error estimate in $\mathcal{W}_2$-distance with a reduced convergence rate. Such a Taylor scheme offers a positive answer to the question **(Q1)**, but at the expense of additional evaluations of two higher-order derivatives of the potential function, i.e., $\nabla^2 U$ and $\nabla(\Delta U)$. Even worse, the presence of $\nabla(\Delta U)$ increases the dimension dependence of the moment bound of the algorithm (7) (see Proposition 3 in Sabanis and Zhang (2019)). In order to remedy it, Li et al. (2019) introduced a stochastic Runge–Kutta method (termed as SRK-LD), originally proposed by Milstein and Tretyakov (2004) for general additive SDEs, for the Langevin dynamics (2) and established a non-asymptotic error bound of $O(d^{3/2}h^{3/2})$ in $\mathcal{W}_2$-distance with a strongly convex potential, considerably reducing the dimension dependence derived in Sabanis and Zhang (2019). The Hessian-free Runge–Kutta (RK) algorithm there requires evaluations of three gradients at each iteration (see the scheme (9) in Li et al. (2019)), which is naturally expected as two gradients are seemingly needed to approximate the Hessian $\nabla^2 U$ and three gradients to approximate the third derivative $\nabla(\Delta U)$ in the Itô Taylor expansions of the Langevin SDE (2). To further save computational costs, we raise the following interesting question:

**(Q2).** *Are there any Runge–Kutta type LMC sampling algorithms involved with only two gradient evaluations at each iteration and what are their non-asymptotic error bounds beyond log-concavity?*

As discussed above, this algorithmic task is not trivial due to the need to properly approximate the third derivative $\nabla(\Delta U)$. In this article, we answer the question **(Q2)** in the affirmative. By incorporating several method parameters, we introduce a family of stochastic Runge–Kutta methods with three stages, formulated by (10)-(11), for the Langevin SDE (2). Several order conditions are provided to ensure the strong convergence rate of order 1.5 in finite time (see (12)-(14)). Solving these conditions results in a unique efficient Runge–Kutta Langevin Monte Carlo (15)-(16) (termed as RKLMC-2G) with only two gradient evaluations (i.e., two stages) at each iteration. Allowing for three gradient evaluations would leave us enough freedom in order conditions to admit infinitely many RKLMC methods with three stages. A rigorous analysis of non-asymptotic error bounds is presented for the general RKLMC methods (10)-(11) including RKLMC-2G as a special case. In particular, a uniform-in-time convergence rate of order $O(d^{\frac{3}{2}}h^{\frac{3}{2}})$ is derived in a log-Sobolev inequality (LSI) setting, coinciding with the convergence rate proved in Li et al. (2019) under the log-concavity condition.

We mention that our uniform-in-time error analysis differs from that in Li et al. (2019), where the contractivity of the mean-square error propagation was available due to the convexity assumption. Instead, we work in a non-convex setting and do not have such contractivity. Inspired by Yang and Wang (2025), our uniform-in-time error analysis mainly consists of two steps. First, via the analysis of the one-step approximation error we obtain the finite-time mean-square error bounds based on the fundamental convergence theorem, but suffering from exponential dependence on the length of time (see Proposition 3.3). Secondly, we combine the finite-time convergence error with uniform-in-time moment bounds of the RKLMC algorithms (Proposition 3.2) and the exponential ergodicity of SDE in the non-convex setting (Proposition 3.4) to arrive at the uniform-in-time convergence error bound (Theorem 2.6).

**Contributions.** The main contribution of this work can be summarized as follows:

- A novel higher-order and Hessian-free LMC sampling algorithm is proposed. Compared with the existing Runge–Kutta LMC (Li et al., 2019) involved with three gradient evaluations, the newly proposed algorithm is computationally cheaper and requires only two gradient evaluations for every iteration.

- Under certain log-smooth conditions and in a non-log-concave (i.e., LSI) setting, we establish non-asymptotic

error bounds in $\mathcal{W}_2$-distance for a general class of Runge–Kutta methods, including the proposed algorithm as a special case. The obtained uniform-in-time convergence rate of order $O(d^{\frac{3}{2}}h^{\frac{3}{2}})$ matches the order obtained in Li et al. (2019) under the log-concavity condition. To the best of our knowledge, this is the first convergence guarantee for higher-order and Hessian-free LMC sampling algorithms beyond log-concavity.

The rest of this paper is organized as follows. The next section introduces the accelerated LMC algorithm and provides main theoretical results for the proposed algorithm. Section 3 presents an outline of the proof of the main result. Numerical experiments are reported in Section 4 and some concluding remarks are given in the last section.

## 2. Main Results

### 2.1. Notation.

Throughout this paper, we use $\mathbb{N}$ to denote the set of all positive integers and let $\mathbb{N}_0 := \mathbb{N} \cup \{0\}$. For any $n \in \mathbb{N}$, let $[n] := \{1, 2, \ldots, n\}$, $[n]_0 := \{0, 1, \ldots, n\}$. The notation $\widetilde{O}(\cdot)$ stands for $O(\cdot) \log^{O(1)}(\cdot)$. Let $\langle \cdot, \cdot \rangle$ and $(\cdot \otimes \cdot)$ denote, respectively, the inner and outer products of vectors in $\mathbb{R}^d$. We denote by $|\cdot|$ the Euclidean norm of vectors in $\mathbb{R}^{d_1}$ and by $\|\cdot\|$ the induced operator norm of matrices in $\mathbb{R}^{d_1 \times d_2}$ with $d_1, d_2 \in \mathbb{N}$.

Let $\mathbb{R}^{d_1 \times \cdots \times d_n}$ be the space of $n$-th order tensors of dimensions $d_1, \ldots, d_n \in \mathbb{N}$. For any tensor $\mathcal{A} = (a_{i_1 \cdots i_n}) \in \mathbb{R}^{d_1 \times \cdots \times d_n}$ and vectors $x_k = (x_{k1}, \ldots, x_{kd_k}) \in \mathbb{R}^{d_k}$ for $k \in [n]$, define the associated multilinear form by

$$\mathcal{A}(x_1, \ldots, x_n) := \sum_{i_1=1}^{d_1} \cdots \sum_{i_n=1}^{d_n} a_{i_1 \cdots i_n} x_{1i_1} \cdots x_{ni_n}.$$

The spectral norm of $\mathcal{A}$ is defined as

$$\|\mathcal{A}\| := \sup_{|x_k|=1,\, k \in [n]} |\mathcal{A}(x_1, \ldots, x_n)|. \tag{4}$$

Let $\mathcal{P}(\mathbb{R}^d)$ denote the space of probability distributions on $\mathbb{R}^d$. For any $\nu_1, \nu_2 \in \mathcal{P}(\mathbb{R}^d)$, denote their $L^p$-Wasserstein ($\mathcal{W}_p$ in short) distance by

$$\mathcal{W}_p(\nu_1, \nu_2) := \inf_{\gamma \in \Gamma(\nu_1, \nu_2)} \left( \int_{\mathbb{R}^d \times \mathbb{R}^d} |x - y|^p \mathrm{d}\gamma(x, y) \right)^{1/p},$$

where $\Gamma(\nu_1, \nu_2)$ denotes the set of probability distributions on $\mathbb{R}^d \times \mathbb{R}^d$ with marginal distributions $\nu_1$ and $\nu_2$. Let $\mathcal{L}(X)$ denote the law of the random variable $X$.

Let $\mathcal{C}_b(\mathbb{R}^d, \mathbb{R})$ and $\mathcal{B}_b(\mathbb{R}^d, \mathbb{R})$ denote the spaces of all bounded continuous functions and bounded Borel measurable functions from $\mathbb{R}^d$ to $\mathbb{R}$, respectively. For any $k \in \mathbb{N}_0$,

we denote by $\mathcal{C}^k(\mathbb{R}^d, \mathbb{R})$ the space of $k$-times continuously differentiable functions from $\mathbb{R}^d$ to $\mathbb{R}$, and by $\mathcal{C}_b^k(\mathbb{R}^d, \mathbb{R})$ the subspace consisting of functions whose derivatives up to order $k$ are bounded and continuous.

For $f \in \mathcal{C}^k(\mathbb{R}^d, \mathbb{R})$, we denote the gradient and the Hessian of $f$ by $\nabla f$ and $\nabla^2 f$, respectively. The Laplacian of $f$ is denoted by $\Delta f$. Moreover, for any $x, v_1, \cdots v_{k-1} \in \mathbb{R}^d$, we define the $k$-th order derivative of $f$ by $\nabla^k f(x)$:

$$\nabla^k f(x)(v_1, \cdots, v_{k-1}) := \nabla \langle \cdots \langle \nabla f(x), v_1 \rangle, \cdots v_{k-1} \rangle, \tag{5}$$

where the $l$-th component for $l \in [d]$ is given by

$$\begin{aligned}
&\left( \nabla^k f(x)(v_1, \cdots v_{k-1}) \right)^{(l)} \\
&:= \sum_{l_1=1}^{d} \cdots \sum_{l_{k-1}=1}^{d} v_{1l_1} \cdots v_{k-1l_{k-1}} \frac{\partial^k f(x)}{\partial x_l \partial x_{l_1} \cdots \partial x_{l_{k-1}}}.
\end{aligned} \tag{6}$$

### 2.2. Accelerated LMC Algorithms

As a classical argument (Kloeden & Platen, 1992; Milstein & Tretyakov, 2004), the Itô Taylor expansions can be employed to construct high-order strong approximation schemes for SDEs, commonly referred to strong Taylor approximations. For instance, based on a truncation of the Itô Taylor expansions up to strong order 1.5, one can construct an order 1.5 strong Taylor scheme for the Langevin SDE (2):

$$\begin{aligned}
\mathbb{Y}_{n+1} =& \mathbb{Y}_n - \nabla U(\mathbb{Y}_n)h + \sqrt{2}\Delta W_{n+1} \\
& + \tfrac{1}{2}\nabla^2 U(\mathbb{Y}_n)\nabla U(\mathbb{Y}_n)h^2 - \tfrac{1}{2}\nabla(\Delta U(\mathbb{Y}_n))h^2 \\
& - \sqrt{2}\nabla^2 U(\mathbb{Y}_n)\Delta Z_{n+1}, \quad n \in \mathbb{N}_0,
\end{aligned} \tag{7}$$

where $\mathbb{Y}_0 = X_0$ and the increments are defined by

$$\Delta W_{n+1} := \int_{t_n}^{t_{n+1}} \mathrm{d}W_t, \ \Delta Z_{n+1} := \int_{t_n}^{t_{n+1}} \int_{t_n}^{t} \mathrm{d}W_s \,\mathrm{d}t. \tag{8}$$

Let $\Delta W_n^k$ and $\Delta Z_n^k$, $k \in [d]$, be the $k$-th components of $\Delta W_n$ and $\Delta Z_n$, respectively. For any $k \in [d]$, the pairs $(\Delta W_n^k, \Delta Z_n^k)_{n \in \mathbb{N}}$ are jointly Gaussian with zero mean, variances

$$\mathbb{E}[|\Delta W_n^k|^2] = h, \quad \mathbb{E}[|\Delta Z_n^k|^2] = \frac{h^3}{3},$$

and covariance

$$\mathbb{E}[\Delta W_n^k \Delta Z_n^k] = \frac{h^2}{2}.$$

In practice, the pairs $(\Delta W_n, \Delta Z_n)_{n \in \mathbb{N}}$ can be generated by two sequences of independent standard Gaussian random variables $\xi_n, \eta_n \sim \mathcal{N}(0, I_d)$ via the linear transformation

$$\Delta W_n = h^{\frac{1}{2}}\xi_n, \quad \Delta Z_n = h^{\frac{3}{2}}\left(\tfrac{1}{2}\xi_n + \tfrac{1}{2\sqrt{3}}\eta_n\right). \tag{9}$$

Although the Taylor scheme (7), termed as Taylor–expanded Langevin Monte Carlo (TELMC) algorithm, attains order

1.5 strong convergence, higher than (1), the algorithm relies on multiple evaluations of higher-order derivatives of the potential function, such as $\nabla^2 U$ and $\nabla(\Delta U)$. This would be computationally expensive especially in high-dimensional settings. Motivated by the idea of Runge–Kutta methods for SDEs (Kloeden & Platen, 1992; Milstein & Tretyakov, 2004; Burrage & Burrage, 2000; Rößler, 2010), we turn to Runge–Kutta Langevin Monte Carlo (RKLMC) algorithms, which achieve the same high-order accuracy while avoiding any higher order derivatives beyond the gradient. We introduce the following general explicit RKLMC schemes with three stages $Y_n$, $\Phi_1^n$ and $\Phi_2^n$, given by $Y_0 = X_0$ and

$$
\begin{aligned}
Y_{n+1} =& Y_n - (1 - \alpha - \beta)\nabla U(Y_n)h - \alpha\nabla U(\Phi_1^n)h \\
& - \beta\nabla U(\Phi_2^n)h + \sqrt{2}\Delta W_{n+1},
\end{aligned}
\tag{10}
$$

where $\alpha, \beta \in \mathbb{R}$, the stages $\Phi_1^n$ and $\Phi_2^n$ are given by

$$
\begin{aligned}
\Phi_1^n =& Y_n - a_{11}\nabla U(Y_n)h + \sqrt{2}b_1\frac{\Delta Z_{n+1}}{h}, \\
\Phi_2^n =& Y_n - a_{21}\nabla U(Y_n)h - a_{22}\nabla U(\Phi_1^n)h + \sqrt{2}b_2\frac{\Delta Z_{n+1}}{h}.
\end{aligned}
\tag{11}
$$

Here, the coefficients of the RKLMC (10)-(11) are required to satisfy the following order conditions:

$$
\begin{aligned}
&(1) \quad \alpha a_{11} + \beta(a_{21} + a_{22}) = \tfrac{1}{2}, &(12) \\
&(2) \quad \alpha b_1 + \beta b_2 = 1, &(13) \\
&(3) \quad \alpha(b_1)^2 + \beta(b_2)^2 = \tfrac{3}{2}. &(14)
\end{aligned}
$$

It is worthwhile to highlight that the above order conditions are obtained by performing the Taylor–expansion of RKLMC scheme (10)-(11) and comparing it with the TELMC method (7). More formally, the coefficients $1-\alpha-\beta, \alpha, \beta$ are chosen so that the gradient weights sum to one, matching the term $\nabla U$ in TELMC (7). The condition (12) arises from matching the term $\nabla^2 U\nabla U$ and the condition (13) is obtained by matching the term $\nabla^2 U\,\Delta Z_{n+1}$. The condition (14) is used to match the quadratic variation term $\nabla(\Delta U(Y_n))$. For details of this construction, please refer to Appendix B.2.

To rely on only two gradient evaluations, we take $a_{11} = b_1 = 0$ so that the stage $\Phi_1^n = Y_n$. Then solving the above order conditions results in a unique efficient RKLMC scheme requiring only two gradient evaluations at each iteration:

$$
Y_{n+1} = Y_n - \tfrac{1}{3}\nabla U(Y_n)h - \tfrac{2}{3}\nabla U(\Phi^n)h + \sqrt{2}\,\Delta W_{n+1},
\tag{15}
$$

where $Y_0 = X_0$ and the stage $\Phi^n$ is defined by

$$
\Phi^n = Y_n - \tfrac{3}{4}\nabla U(Y_n)h + \tfrac{3\sqrt{2}\Delta Z_{n+1}}{2h}.
\tag{16}
$$

We would like to mention that the proposed RKLMC (15)-(16) (termed as RKLMC-2G for short) requires only 2 evaluations of $\nabla U$ per step, which is fewer than the 3 evaluations needed by the SRK-LD scheme considered in Li et al.

(2019). This reduction would lead to a considerable saving in computational costs.

Allowing for three gradient evaluations would leave us enough freedom in order conditions to admit infinitely many RK methods with three stages (termed as RKLMC-3G). As a particular example, we solve the above order conditions and list a RK method with three gradients as follows:

$$
\begin{aligned}
Y_{n+1} =& Y_n - \tfrac{1}{4}\nabla U(Y_n)h - \tfrac{1}{4}\nabla U(\Phi_1^n)h \\
& - \tfrac{1}{2}\nabla U(\Phi_2^n)h + \sqrt{2}\Delta W_{n+1},
\end{aligned}
\tag{17}
$$

where the stages $\Phi_1^n$ and $\Phi_2^n$ are given by

$$
\begin{aligned}
\Phi_1^n =& Y_n + 2\sqrt{2}\tfrac{\Delta Z_{n+1}}{h}, \\
\Phi_2^n =& Y_n - \tfrac{1}{2}\nabla U(Y_n)h - \tfrac{1}{2}\nabla U(\Phi_1^n)h + \sqrt{2}\tfrac{\Delta Z_{n+1}}{h}.
\end{aligned}
\tag{18}
$$

Another example of RKLMC-3G is

$$
Y_{n+1} = Y_n - \tfrac{2}{3}\nabla U(\Phi_1^n)h - \tfrac{1}{3}\nabla U(\Phi_2^n)h + \sqrt{2}\Delta W_{n+1},
\tag{19}
$$

where the stages $\Phi_1^n$ and $\Phi_2^n$ are given by

$$
\begin{aligned}
\Phi_1^n =& Y_n - \tfrac{1}{2}\nabla U(Y_n)h + \tfrac{\sqrt{2}}{2}\tfrac{\Delta Z_{n+1}}{h}, \\
\Phi_2^n =& Y_n - \tfrac{1}{2}\nabla U(Y_n)h + 2\sqrt{2}\tfrac{\Delta Z_{n+1}}{h}.
\end{aligned}
\tag{20}
$$

In numerically solving ordinary differential equations (ODEs) via RK methods, it is common to find optimal method parameters for RK applied to linear ODEs by minimizing the leading local truncation error constants (Hairer et al., 1993). A similar idea can be also applied here for the above SRK methods with three gradients, to possibly derive best ones. This is left for our future study.

### 2.3. Main Results

This subsection presents our main results on accelerated LMC algorithms (10)-(11). We first delineate necessary assumptions underlying our results. Let us begin with two standard conditions in the literature.

**Assumption 2.1** (Dissipativity condition). There exist two constants $\mu, \mu' > 0$ such that

$$
\langle x, \nabla U(x)\rangle \geq \mu|x|^2 - \mu'd, \quad \forall x \in \mathbb{R}^d.
\tag{21}
$$

Assumption 2.1 is commonly imposed in the non-asymptotic error analysis of LMC algorithms (Mou et al., 2022; Pang et al., 2025; Erdogdu & Hosseinzadeh, 2021), which ensures the uniform boundedness of moments of LMC algorithms as well as the continuous-time Langevin dynamics.

**Assumption 2.2** (Gradient Lipschitz condition). Assume that the potential $U \in \mathcal{C}^2(\mathbb{R}^d, \mathbb{R})$ and there exists a constant $L_1 > 0$ such that

$$
\big|\nabla U(x) - \nabla U(y)\big| \leq L_1|x - y|, \quad \forall x, y \in \mathbb{R}^d.
\tag{22}
$$

Assumption 2.2 immediately implies a linear growth condition on the gradient $U$. More precisely, there exists a constant $L_1' > 0$ such that, for any $x \in \mathbb{R}^d$,

$$|\nabla U(x)| \le |\nabla U(0)| + L_1|x| \le L_1' d^{\frac{1}{2}} + L_1|x|. \quad (23)$$

For higher-order LMC algorithms, it is standard to impose stronger smoothness assumptions on the potential function $U$, involving higher-order derivatives (Li et al., 2019; Sabanis & Zhang, 2019).

**Assumption 2.3** (Hessian Lipschitz condition). Assume the potential $U \in \mathcal{C}^3(\mathbb{R}^d, \mathbb{R})$ and there exists a constant $L_2 > 0$ such that

$$\|\nabla^2 U(x) - \nabla^2 U(y)\| \le L_2|x - y|, \quad \forall x, y \in \mathbb{R}^d. \quad (24)$$

**Assumption 2.4** (3rd-order derivative Lipschitz condition). Assume the potential $U \in \mathcal{C}^4(\mathbb{R}^d, \mathbb{R})$ and there exists a constant $L_3 > 0$ such that

$$\|\nabla^3 U(x) - \nabla^3 U(y)\| \le L_3|x - y|, \quad \forall x, y \in \mathbb{R}^d. \quad (25)$$

We would like to emphasize that all constants used here $(\mu, \mu', L_1, L_1', L_2, L_3)$ are of constant order, independent of the problem dimension $d$. Also, we note that such smoothness assumptions up to 3rd-order derivatives are standard ones in analyzing higher-order Langevin Monte Carlo methods; see, e.g., (Li et al., 2019; Sabanis & Zhang, 2019).

Let $\{p_t\}_{t \ge 0}$ and $\{q_n\}_{n \in \mathbb{N}_0}$ denote the Markov semigroups associated with the solutions of the Langevin SDE (2) and the RKLMC algorithm (10)-(11), respectively.

**Assumption 2.5** (Log-Sobolev inequality). Let the target distribution $\pi(\mathrm{d}x) \propto e^{-U(x)}\mathrm{d}x$ satisfy the log-Sobolev inequality with constant $\rho$:

$$\pi(\phi^2 \log \phi^2) \le \rho\pi(|\nabla\phi|^2), \forall \phi \in C_b^1(\mathbb{R}^d), \pi(\phi^2) = 1, \quad (26)$$

where the constant $\rho$ does not depend on $d$.

The LSI is only used to guarantee exponential ergodicity in $\mathcal{W}_2$-distance of the Langevin dynamics (see Proposition 3.4). In Section 4, we will provide some concrete examples satisfying all the required assumptions. Under the above assumptions, we can now state the main result of this paper.

**Theorem 2.6** (Main results for RKLMC). *Let Assumptions 2.1-2.5 be fulfilled and let*

$$\kappa_1 := 4\alpha^2(a_{11})^2 + \beta^2\big(6(a_{21})^2 + 18(a_{22})^2 + 9(a_{11}a_{22})^2\big).$$

*Assume that, for any $q \le 3$, there exists a dimension-independent constant $\sigma(q) > 0$ such that the initial value obeys*

$$\mathbb{E}[|X_0|^{2q}] \le \sigma(q)d^q.$$

*If the uniform stepsize $h$ satisfies*

$$h \le 1 \wedge \tfrac{1}{2L_1'} \wedge \tfrac{1}{2L_1} \wedge \tfrac{4}{\mu} \wedge \tfrac{\mu}{32L_1^2} \wedge \tfrac{\mu}{4\kappa_1 L_1^2} \wedge \tfrac{\mu^2}{8\kappa_1 L_1^3},$$

*then for any $n \in \mathbb{N}$ and any initial distribution $\nu := \mathcal{L}(X_0)$, there exist two dimension-independent constants $C_1, C_2$ such that*

$$\mathcal{W}_2(\nu q_n, \pi) \le C_1 d^{\frac{3}{2}} h^{\frac{3}{2}} + C_2 d^{\frac{1}{2}} e^{-\lambda nh}, \quad (27)$$

*where $\lambda := \frac{\eta}{\log \mathcal{K} + 1 + \eta/(2L_1)}$.*

As a direct consequence of this theorem, we have the following result on the mixing time.

**Proposition 2.7** (Mixing time for RKLMC). *Let all conditions in Theorem 2.6 be satisfied. Then, to achieve a prescribed accuracy level $\epsilon > 0$ in $\mathcal{W}_2$-distance, the number of iterations required for RKLMC (10) is of order $\widetilde{O}(d\epsilon^{-\frac{2}{3}})$.*

The proofs of Theorem 2.6 and Proposition 2.7 are deferred to Appendix C. In Table 1, we compare the number of iterations of three accelerated LMC algorithms required to achieve $\epsilon$ error in $\mathcal{W}_2$-distance. It is indicated that, in the strongly log-concave case, the RKLMC algorithm attains sharper error bounds than the TELMC scheme. Moreover, our RKLMC method requires fewer gradient evaluations than the RK algorithm proposed in Li et al. (2019) and improves upon the best-known convergence rates in non-convex settings.

*Table 1.* Comparison of accelerated LMC sampling algorithms.

| R/A | H-D | S-C | N-G | M-T |
|---|---|---|---|---|
| PAPER[A] / TELMC | YES | YES | NA | $\widetilde{O}(d^{\frac{4}{3}}\epsilon^{-\frac{2}{3}})$ |
| PAPER[B] / TELMC | YES | NO | NA | $\widetilde{O}(e^{O(d)}\epsilon^{-\frac{4}{3}})$ |
| PAPER[C] / RKLMC | NO | YES | 3 | $\widetilde{O}(d\epsilon^{-\frac{2}{3}})$ |
| THIS WORK / RKLMC | NO | NO | 2 | $\widetilde{O}(d\epsilon^{-\frac{2}{3}})$ |

R/A: Reference/Algorithm
H-D: Higher-order Derivatives beyond the Gradient
S-C: Strong Convexity
N-G: Number of gradient evaluations for one iteration
M-T: Mixing Time
NA: Not Applicable
[A] Sabanis and Zhang (2019)
[B] Neufeld and Zhang (2024)
[C] Li et al. (2019)

## 3. Overview of Non-asymptotic Error Analysis

In this section we present an overview of the non-asymptotic error analysis of the RKLMC algorithm (10)-(11).

The objective of Theorem 2.6 is to derive a bound for $\mathcal{W}_2(\nu q_n, \pi)$. Applying the triangle inequality and consider-

ing a fixed time $T := n_1 h$, we have

$$\mathcal{W}_2(\nu q_n, \pi) \leq \underbrace{\mathcal{W}_2(\nu q_{n-n_1} q_{n_1}, \nu q_{n-n_1} p_T)}_{\text{Finite-time error}}$$
$$+ \underbrace{\mathcal{W}_2(\nu q_{n-n_1} p_T, \pi)}_{\text{Exponential ergodicity}}, \quad n \geq n_1. \quad (28)$$

Thanks to this decomposition, the proof of Theorem 2.6 can be divided into four main parts, as outlined below.

**Part 1: Uniform-in-Time Moment Bounds.**

We first establish uniform-in-time moment bounds for both the Langevin SDE (2) and the RKLMC algorithm (10)-(11) by leveraging the dissipativity assumption.

**Proposition 3.1** (Uniform-in-time moment bounds of Langevin dynamics). *Let Assumption 2.1 hold. Then for $p \geq 1$, there exists a constant $c \in (0, 2\mu)$ such that*

$$\mathbb{E}\big[|X_t|^{2p}\big] \leq e^{-cpt}\mathbb{E}\big[|X_0|^{2p}\big] + \mathcal{M}_1(p)d^p, \quad (29)$$

*where*

$$\mathcal{M}_1(p) := \frac{(4p-2+2\mu')^p}{p}\Big(\frac{p-1}{(2\mu-c)p}\Big)^{p-1}. \quad (30)$$

Such estimates have been established previously (see, e.g., Lemma 2.4 of Yang and Wang (2025) and Lemma 3.1 of Pang et al. (2025)). We now present uniform-in-time moment bounds for RKLMC, with the detailed proof provided in Appendix A.

**Proposition 3.2** (Uniform-in-time moment bounds for RKLMC). *Let Assumption 2.1, 2.2 hold. Assume that the uniform timestep $h$ satisfies the condition in Theorem 2.6. Then for any $p \geq 1$, it holds*

$$\mathbb{E}[|Y_n|^{2p}] \leq e^{-\frac{\mu}{4}t_n}\mathbb{E}[|X_0|^{2p}] + \mathcal{M}_2(p)d^p, \quad (31)$$

*where $\mathcal{M}_2(p) = C(\mu, \mu', p, \alpha, \beta, a_{11}, a_{21}, a_{22}, b_1, b_2, L_1, L_1')$ is independent of $d$ and $Y_n$ is produced by (10).*

**Part 2: Finite-Time Strong Error Bound.**

Armed with the above uniform moment bound, we first analyze the finite-time strong approximation error between RKLMC (10)-(11) and the Langevin dynamics (2).

**Proposition 3.3** (Error analysis of RKLMC in finite time). *Let Assumptions 2.1- 2.4 hold. Suppose that the uniform stepsize $h$ satisfies the condition in Theorem 2.6. Then for fixed $T = n_1 h$, $n_1 \in \mathbb{N}$, it holds*

$$\sup_{n\in[n_1]} \mathbb{E}\big[\big|X_{t_n} - Y_n\big|^2\big] \leq C(T)\big(K_1 d^3 + K_2\mathbb{E}[|X_0|^6]\big)h^3, \quad (32)$$

*where*

$$C(T) = e^{(1+12L_1)T}, \quad (33)$$

*$K_1$ and $K_2$ are two dimension-independent constants, depending on $\mu, \mu', \alpha, \beta, a_{11}, a_{21}, a_{22}, b_1, b_2, L_1, L_1', L_2, L_3$.*

The detailed proof of this proposition is presented in Appendix B. From this, the first term on the right-hand side of (28) can be bounded explicitly in terms of $T$:

$$\mathcal{W}_2(\nu q_{n-n_1} q_{n_1}, \nu q_{n-n_1} p_T) \leq C(T)h^{\frac{3}{2}}. \quad (34)$$

**Part 3: Exponential Ergodicity of Langevin Dynamics.**

To control the second term in (28), we exploit the exponential ergodicity of the Langevin semigroup $\{p_t\}_{t\geq 0}$.

**Proposition 3.4** (Exponential ergodicity in $\mathcal{W}_2$-distance). *Let Assumptions 2.2 and 2.5 hold. Then there exist two constants $\mathcal{K}, \eta > 0$, independent of $d, t$, such that*

$$\mathcal{W}_2(\nu p_t, \pi) \leq \mathcal{K}e^{-\eta t}\mathcal{W}_2(\nu, \pi), \quad (35)$$

*for any $t \geq 0$ and any initial distribution $\nu := \mathcal{L}(X_0)$.*

This proposition is quoted from Theorem 2.1(2) and 2.6(2) of Wang (2020) and Proposition 2.5 of Yang and Wang (2025).

Consequently, the second term of (28) satisfies

$$\mathcal{W}_2(\nu q_{n-n_1} p_T, \pi) \leq \mathcal{K}e^{-\eta T}\mathcal{W}_2(\nu q_{n-n_1}, \pi), \quad (36)$$

which is essential for the uniform-in-time error analysis of the RKLMC algorithm.

**Part 4: Uniform-in-Time Error Bound in $\mathcal{W}_2$-Distance.**

Finally, we combine (34) and (36) and choose $T = \Theta$ such that $\mathcal{K}e^{-\eta T} = 1/e$ to obtain

$$\mathcal{W}_2(\nu\tilde{p}_n, \pi) \leq C(\Theta)h^{\frac{3}{2}} + \frac{1}{e}\mathcal{W}_2(\nu q_{n-n_1}, \pi). \quad (37)$$

Iterating this inequality yields

$$\mathcal{W}_2(\nu q_n, \pi) \leq C_1 h^{\frac{3}{2}} + C_2 e^{-\lambda n h}, \quad (38)$$

as desired. For more details, see the proof of Theorem 2.6 in Appendix C.

## 4. Numerical Experiments

In this section, we present several numerical experiments for the Gaussian mixture model (GMM) and Bayesian logistic regression (BLR) to validate the above theoretical findings.

### 4.1. Convergence Rate and Dimension Dependence

**1). Two-mode Gaussian Mixture Model:** We consider a two-component Gaussian mixture target with the potential

$$U_1(x) = -\log\big(\tfrac{1}{2}e^{-\frac{1}{2}|x-\mu_1|^2} + \tfrac{1}{2}e^{-\frac{1}{2}|x-\mu_2|^2}\big),$$

where the mean vector is chosen as $\mu_1 = \frac{2}{\sqrt{d}}(1, \cdots, 1)^T \in \mathbb{R}^d$ and $\mu_2 = -\mu_1$, ensuring that $|\mu_1| = |\mu_2| = 2$.

This potential is non-convex whenever $\|\mu_i\| \geq 1, i = 1, 2$ (Dalalyan, 2017b). As verified in Li et al. (2019), the 1-st to 3-rd-order Lipschitz conditions are fulfilled. Moreover, the log-Sobolev inequality and the dissipativity condition were established in Yang and Wang (2025). Therefore, all assumptions required in this work are satisfied for this example.

**2). Bayesian Logistic Regression:** We next consider the posterior distribution of a Bayesian logistic regression model with the potential

$$U_2(\theta) = -Y^\top X\theta + \sum_{i=1}^{n} \log\big(1 + \exp(\theta^\top x_i)\big) + \frac{\alpha}{2}\big\|\Sigma_X^{1/2}\theta\big\|_2^2.$$

The model is constructed from synthetic data and the prior strength is fixed as $\alpha = 0.5$. For a given dimension $d$, the true parameter is taken to be $\theta_{true} = \frac{1}{\sqrt{d}}\mathbf{1}_d$. The design matrix $X \in \mathbb{R}^{n \times d}$ is generated with i.i.d. standard Gaussian entries, where $n = 100$. The response variables are then sampled from the logistic model with success probabilities $p_i = 1/(1 + \exp(-x_i^\top \theta_{true}))$, namely, $Y_i \sim \text{Bernoulli}(p_i)$. We further define the empirical covariance matrix by $\Sigma_X = \frac{X^\top X}{n}$.

We note that $U_2$ is strongly convex provided that $\Sigma_X$ is positive definite (Li et al., 2019; Dalalyan, 2017b). Consequently, $U_2$ satisfies both the log-Sobolev inequality and the dissipativity condition. Moreover, since the derivatives of the logistic loss are bounded and the prior term is quadratic, $U_2$ has globally Lipschitz derivatives up to order three, with constants depending on $X$. Hence, all assumptions required in this work are satisfied.

To examine the convergence rate, we simulate the LMC, SRK-LD and RKLMC-2G algorithms for the above two examples. For SRK-LD and RKLMC-2G algorithms, we also examine the dimension dependence. All schemes start from the same initial condition $X_0 = 0$ and are simulated up to the terminal time $T = 2$. In each experiment, we generate $M = 5000$ independent trajectories. To ensure a fair comparison, all coarse-grid approximations are constructed from a common Brownian path generated on a sufficiently fine reference grid with stepsize $h_{\mathrm{ref}}$. The reference solution is taken to be the finest-grid LMC trajectory. The considered error is the time discretization error, measured by root mean-square error $(\mathbb{E}|Y_{N_T}^h - X_T^{\mathrm{ref}}|^2)^{1/2}$, where $Y_{N_T}^h$ denotes the terminal value of the numerical scheme with stepsize $h$ and $X_T^{\mathrm{ref}}$ denotes the reference solution at the terminal time $T$. Such a mean-square error is examined here since the proposed stochastic Runge-Kutta schemes are constructed and analyzed via their mean-square convergence rate analysis and our non-asymptotic error bounds in $\mathcal{W}_2$-distance also rely on the finite-time mean-square errors.

To test the convergence rate, we fix the dimension $d = 10$,

take $h_{\mathrm{ref}} = 2^{-15}$ and compute the coarse approximations with stepsizes $h = 2^{-10}, \dots, 2^{-6}$. Root mean-square errors are then plotted against stepsizes on a log-log scale.

To test the dimension dependence, we vary the dimension over $d = 8, 10, 12, 14, 16$ for GMM and over $d = 6, 8, 10, 12, 14$ for BLR. The reference stepsize is chosen as $h_{\mathrm{ref}} = 2^{-9}$ for GMM and $h_{\mathrm{ref}} = 2^{-11}$ for BLR and the coarse approximations are computed with the fixed stepsizes $h = 2^{-4}$ and $h = 2^{-6}$, respectively. Root mean-square errors are then plotted against the dimension on a log-log scale.

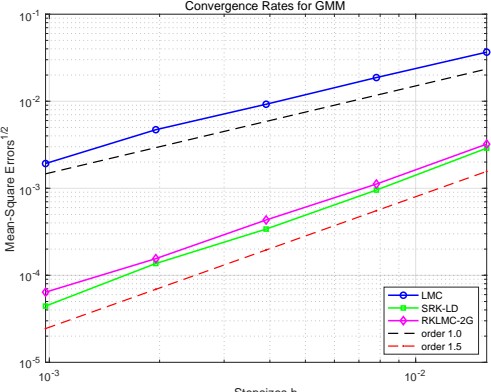

*Figure 1.* Convergence rates of LMC, SRK-LD and RKLMC-2G for GMM.

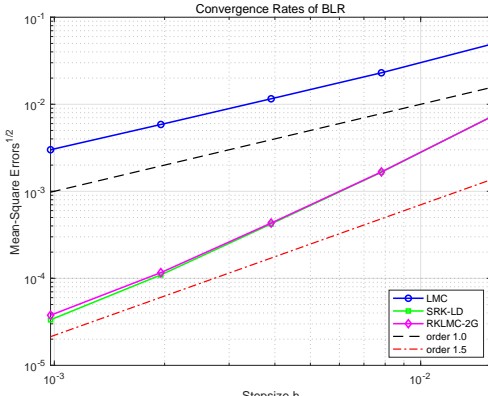

*Figure 2.* Convergence rates of LMC, SRK-LD and RKLMC-2G for BLR.

As shown in Figures 1 and 2, for both GMM and BLR, LMC achieves an empirical convergence rate close to 1, while SRK-LD and RKLMC-2G both attain convergence rates close to 1.5. Also, it is clearly observed that the Runge–Kutta-type algorithms achieve much smaller mean-square errors than LMC of Euler type. In terms of mean-square errors, the difference between RKLMC-2G and SRK-LD is negligible under the same stepsize, while RKLMC-2G reduces the number of gradient evaluations by one-third compared to SRK-LD. In Figures 3 and 4, the slopes with respect to the dimension further indicate that, for these two

models, the errors of SRK-LD and RKLMC-2G scale approximately like $d^{1.5}$. Both the observed convergence rates and dimension dependence agree with our theoretical results.

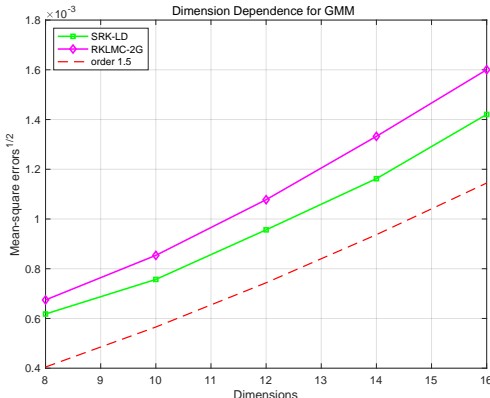

*Figure 3.* Dimension dependence of SRK-LD and RKLMC-2G for GMM.

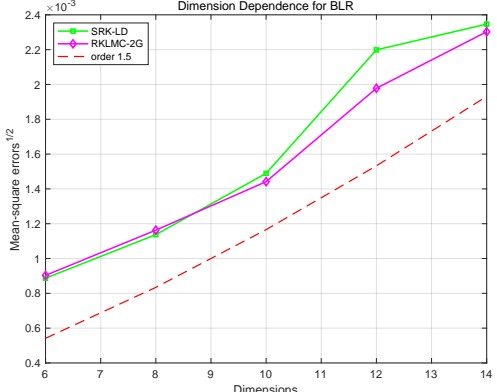

*Figure 4.* Dimension dependence of SRK-LD and RKLMC-2G for BLR.

### 4.2. Sampling Performance

In this subsection, we present several experiments to illustrate the sampling performance of the proposed algorithm.

As the first experiment, we apply the RKLMC-2G scheme to the Langevin diffusion associated with the two-mode GMM. The scheme is initialized at $X_0 = 0$ and simulated up to terminal time $T = 5$. We generate $M = 5000$ independent trajectories on a fine grid with stepsize $h = 2^{-14}$. At time $T = 5$, we record the first component of the terminal samples, plot its empirical histogram and compare it with the exact one-dimensional marginal density of the target GMM. Figure 5 shows that the empirical distribution generated by RKLMC-2G agrees well with the exact marginal density.

As the second experiment, we illustrate the sampling performance of LMC, SRK-LD and RKLMC-2G on a standard eight-mode GMM (Grenioux et al., 2024) by means of two-dimensional scatter plots. The target distribution is an

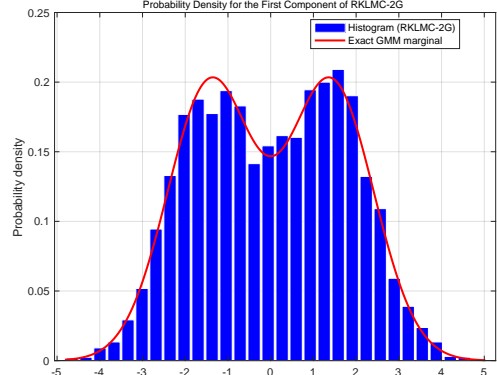

*Figure 5.* Histogram of the first component for RKLMC-2G on the two-mode GMM.

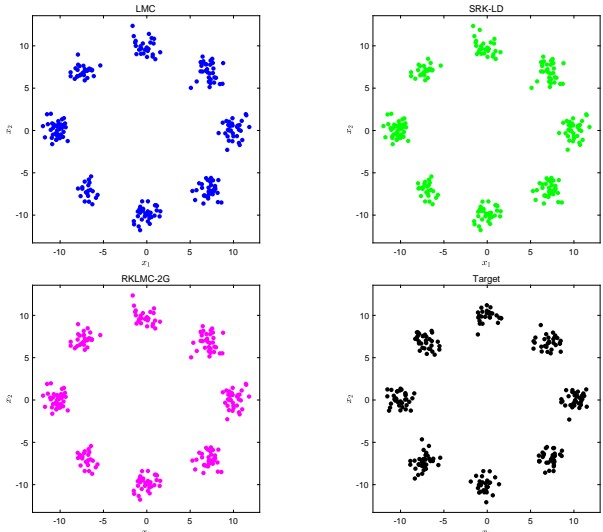

*Figure 6.* Scatter plots for LMC, SRK-LD and RKLMC-2G on the eight-mode GMM.

equally weighted Gaussian mixture in $\mathbb{R}^2$ with density

$$\pi(x) = \frac{1}{8} \sum_{i=0}^{7} \mathcal{N}(x; m_i, 0.7 I_2),$$

where $m_i = 10\big(\cos(2\pi i/8), \sin(2\pi i/8)\big)$, $i = 0, \dots, 7$. We simulate the LMC, SRK-LD, and RKLMC-2G schemes up to terminal time $T = 6$ with stepsize $h = 0.02$, starting from the initial distribution $\mathcal{N}(0, I_2)$. For each method, we generate 256 samples and record their terminal values; for comparison, we also generate 256 independent samples directly from the target distribution. Figure 6 shows that all three methods are able to capture the overall eight-mode structure of the target distribution.

## 5. Conclusion and Future Work

In the present work, we propose a class of Runge–Kutta LMC algorithms, including a particular one with only two gradient evaluations at each iteration. Moreover, under certain non-log-concavity condition, we obtain the non-asymptotic error bound in the $\mathcal{W}_2$-distance of order $O(d^{3/2}h^{3/2})$. In future work, (i) we plan to work on Runge–Kutta type schemes for the underdamped Langevin Monte Carlo; (ii) we also plan to design modified variants of accelerated RKLMC methods to handle the sampling with potentials that may exhibit superlinear growth (Sabanis & Zhang, 2019; Neufeld & Zhang, 2024).

## Acknowledgements

The authors thank Chenxu Pang as well as the anonymous reviewers, the area chair and the program chair for their valuable suggestions, which have significantly improved the quality of this paper. This work was supported by Natural Science Foundation of China (Nos. 12471394, 12371417).

## Impact Statement

This paper presents work whose goal is to advance the field of Machine Learning. There are many potential societal consequences of our work, none which we feel must be specifically highlighted here.

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

## A. Proof of Proposition 3.2

In this section, we establish uniform-in-time moment bounds for the RKLMC algorithm. Prior to proving the proposition, we first state several direct consequences of Assumptions 2.2-2.4, whose proofs rely on the following lemmas.

**Lemma A.1.** *Assume $f \in \mathcal{C}^{k+1}(\mathbb{R}^d, \mathbb{R})$ and its $k$-th derivative satisfies the Lipschitz condition:*

$$\left\| \nabla^k f(x) - \nabla^k f(y) \right\| \le L|x - y|, \quad L > 0, \quad \forall\, x, y \in \mathbb{R}^d. \tag{39}$$

*Then for any $x, v_1, \cdots, v_k \in \mathbb{R}^d$, it holds*

$$\left| \nabla^{k+1} f(x)(v_1, \cdots v_k) \right| \le L|v_1| \cdots |v_k|. \tag{40}$$

*Proof.* For any unit vector $u \in \mathbb{R}^d$, we obtain

$$
\begin{aligned}
\left\langle \nabla^{k+1} f(x)(v_1, \cdots v_k), u \right\rangle &= \lim_{t \to 0} \frac{\nabla^k f(x + tu)(v_1, \cdots v_k) - \nabla^k f(x)(v_1, \cdots v_k)}{t} \\
&= \lim_{t \to 0} \frac{\left(\nabla^k f(x + tu) - \nabla^k f(x)\right)(v_1, \cdots v_k)}{t}.
\end{aligned}
\tag{41}
$$

By choosing $u = \frac{\nabla^{k+1} f(x)(v_1, \cdots v_k)}{|\nabla^{k+1} f(x)(v_1, \cdots v_k)|}$, it follows from the $k$-th derivative Lipschitz condition (39) that

$$\left| \nabla^{k+1} f(x)(v_1, \cdots v_k) \right| \le L \left\| (v_1, \cdots v_k) \right\| \le L|v_1| \cdots |v_k|, \tag{42}$$

where we have used $\|(v_1, \cdots v_k)\| = \|v_1 \otimes \cdots \otimes v_k\| = |v_1| \cdots |v_k|$ in the last step. The proof is thus completed. $\qquad\square$

Owing to this lemma, Assumptions 2.2-2.4 immediately imply that, for any $x, y_1, y_2, y_3 \in \mathbb{R}^d$,

$$|\nabla^2 U(x) y_1| \le L_1 |y_1|, \tag{43}$$

$$|\nabla^3 U(x)(y_1, y_2)| \le L_2 |y_1||y_2|, \tag{44}$$

$$|\nabla^4 U(x)(y_1, y_2, y_3)| \le L_3 |y_1||y_2||y_3|. \tag{45}$$

Additionally, under Assumption 2.3, we claim

$$|\nabla(\Delta U(x))| \le L_2 d, \quad \forall\, x \in \mathbb{R}^d. \tag{46}$$

Now, we aim to check this assertion. Let $h(\cdot) := \Delta U(\cdot)$. By the property of directional derivative, it follows that, for any $x \in \mathbb{R}^d$ and for any unit vector $u \in \mathbb{R}^d$

$$\langle \nabla h(x), u \rangle = \lim_{t \to 0} \frac{h(x + tu) - h(x)}{t}. \tag{47}$$

Noting that $|\operatorname{trace}(M)| \le d\|M\|$, for any $M \in \mathbb{R}^{d \times d}$, one can derive from Assumption 2.3 that

$$
\begin{aligned}
|h(x + tu) - h(x)| &= |\operatorname{trace}(\nabla^2 U(x + tu) - \nabla^2 U(x))| \\
&\le d\|\nabla^2 U(x + tu) - \nabla^2 U(x)\| \\
&\le L_2 t d.
\end{aligned}
\tag{48}
$$

Collecting the above estimate and choosing $u = \frac{\nabla h(x)}{|\nabla h(x)|}$ yields

$$|\nabla(\Delta U(x))| = |\nabla h(x)| = |\langle \nabla h(x), u \rangle| \le L_2 d. \tag{49}$$

Moreover, as indicated by Lemma 35 of (Li et al., 2019), Assumption 2.4 implies

$$|\nabla(\Delta U(x)) - \nabla(\Delta U(y))| \le L_3 d|x - y|, \quad \forall\, x, y \in \mathbb{R}^d. \tag{50}$$

*Proof of Proposition 3.2.* First, one can recast RKLMC (10) as

$$Y_{n+1} = Y_n - \nabla U(Y_n)h + \sqrt{2}\Delta W_{n+1} + \alpha(\nabla U(Y_n) - \nabla U(\Phi_1^n))h + \beta(\nabla U(Y_n) - \nabla U(\Phi_2^n))h. \tag{51}$$

Denote further

$$\delta(\nabla U)_n^{\Phi,i} := \nabla U(Y_n) - \nabla U(\Phi_i^n), \quad i = 1, 2. \tag{52}$$

Taking the squared Euclidean norm on both sides and expanding the resulting terms, we obtain

$$\begin{aligned}
\big|Y_{n+1}\big|^2 =& \big|Y_n\big|^2 + h^2\big|\nabla U(Y_n)\big|^2 + 2\big|\Delta W_{n+1}\big|^2 + \alpha^2 h^2\big|\delta(\nabla U)_n^{\Phi,1}\big|^2 + \beta^2 h^2\big|\delta(\nabla U)_n^{\Phi,2}\big|^2 \\
& - 2h\langle Y_n, \nabla U(Y_n)\rangle + 2\sqrt{2}\langle Y_n, \Delta W_{n+1}\rangle + 2\alpha h\langle Y_n, \delta(\nabla U)_n^{\Phi,1}\rangle + 2\beta h\langle Y_n, \delta(\nabla U)_n^{\Phi,2}\rangle \\
& - 2\sqrt{2}h\langle \nabla U(Y_n), \Delta W_{n+1}\rangle - 2\alpha h^2\langle \nabla U(Y_n), \delta(\nabla U)_n^{\Phi,1}\rangle - 2\beta h^2\langle \nabla U(Y_n), \delta(\nabla U)_n^{\Phi,2}\rangle \\
& + 2\sqrt{2}\alpha h\langle \Delta W_{n+1}, \delta(\nabla U)_n^{\Phi,1}\rangle + 2\sqrt{2}\beta h\langle \Delta W_{n+1}, \delta(\nabla U)_n^{\Phi,2}\rangle + 2\alpha\beta h^2\langle \delta(\nabla U)_n^{\Phi,1}, \delta(\nabla U)_n^{\Phi,2}\rangle.
\end{aligned} \tag{53}$$

Using the Cauchy-Schwarz inequality, the inequality

$$\Big(\sum_{i=1}^k |u_i|\Big)^q \le k^{q-1}\sum_{i=1}^k |u_i|^q, \quad u_i \in \mathbb{R}, \ \forall q, k \in \mathbb{N}, \tag{54}$$

together with the dissipativity condition (21), the gradient Lipschitz condition (22), the linear growth condition (23) and assuming $h \le 1 \wedge \frac{1}{2L_1} \wedge \frac{1}{2L_1'} \wedge \frac{\mu}{32L_1^2}$, it follows that

$$\begin{aligned}
\big|Y_{n+1}\big|^2 \le& \big(1 - \tfrac{3}{2}\mu h\big)\big|Y_n\big|^2 + 4h^2\big|\nabla U(Y_n)\big|^2 + 6\big|\Delta W_{n+1}\big|^2 + 4\alpha^2 h^2\big|\delta(\nabla U)_n^{\Phi,1}\big|^2 + 4\beta^2 h^2\big|\delta(\nabla U)_n^{\Phi,2}\big|^2 \\
& 2\sqrt{2}\langle Y_n, \Delta W_{n+1}\rangle + \tfrac{4}{\mu}\alpha^2 h\big|\delta(\nabla U)_n^{\Phi,1}\big|^2 + \tfrac{4}{\mu}\beta^2 h\big|\delta(\nabla U)_n^{\Phi,2}\big|^2 + 2\mu' dh \\
\le& \big(1 - \tfrac{3}{2}\mu h\big)\big|Y_n\big|^2 + 8L_1^2 h^2\big|Y_n\big|^2 + 6\big|\Delta W_{n+1}\big|^2 + 4\alpha^2 L_1^2 h^2\big|\Phi_1^n - Y_n\big|^2 + 4\beta^2 L_1^2 h^2\big|\Phi_2^n - Y_n\big|^2 \\
& 2\sqrt{2}\langle Y_n, \Delta W_{n+1}\rangle + \tfrac{4}{\mu}\alpha^2 L_1^2 h\big|\Phi_1^n - Y_n\big|^2 + \tfrac{4}{\mu}\beta^2 L_1^2 h\big|\Phi_2^n - Y_n\big|^2 + 8L_1^2 h^2 d + 2\mu' dh \\
\le& \big(1 - \tfrac{5}{4}\mu h\big)\big|Y_n\big|^2 + 6\big|\Delta W_{n+1}\big|^2 + 2\sqrt{2}\langle Y_n, \Delta W_{n+1}\rangle + \underbrace{\alpha^2\big|\Phi_1^n - Y_n\big|^2 + \beta^2\big|\Phi_2^n - Y_n\big|^2}_{=:I_1} \\
& + \underbrace{\tfrac{2}{\mu}\alpha^2 L_1\big|\Phi_1^n - Y_n\big|^2 + \tfrac{2}{\mu}\beta^2 L_1\big|\Phi_2^n - Y_n\big|^2}_{=:I_2} + 4L_1' dh + 2\mu' dh.
\end{aligned} \tag{55}$$

Next, we estimate terms $I_1, I_2$ separately. Recalling (16) and using the linear growth condition (23), (54) as well as $h \le 1 \wedge \frac{1}{2L_1} \wedge \frac{1}{2L_1'}$, one arrives at

$$\begin{aligned}
\big|\Phi_1^n - Y_n\big|^2 \le& 2(a_{11})^2 h^2\big|\nabla U(Y_n)\big|^2 + 4(b_1)^2\big|\tfrac{\Delta Z_{n+1}}{h}\big|^2 \\
\le& 4(a_{11})^2 L_1^2 h^2\big|Y_n\big|^2 + 4(b_1)^2\big|\tfrac{\Delta Z_{n+1}}{h}\big|^2 + 2(a_{11})^2 L_1' dh.
\end{aligned} \tag{56}$$

Before estimating $|\Phi_2^n - Y_n|^2$, we first use the linear growth condition (23), (54) and $h \le 1 \wedge \frac{1}{2L_1} \wedge \frac{1}{2L_1'}$ to show

$$\begin{aligned}
\big|\Phi_1^n\big|^2 \le& 3\big|Y_n\big|^2 + 3(a_{11})^2 h^2\big|\nabla U(Y_n)\big|^2 + 6(b_1)^2\big|\tfrac{\Delta Z_{n+1}}{h}\big|^2 \\
\le& 3\big|Y_n\big|^2 + 6(a_{11})^2 L_1^2 h^2\big|Y_n\big|^2 + 6(b_1)^2\big|\tfrac{\Delta Z_{n+1}}{h}\big|^2 + 6(a_{11})^2 L_1'^2 dh^2 \\
\le& \big(3 + \tfrac{3}{2}(a_{11})^2\big)\big|Y_n\big|^2 + 6(b_1)^2\big|\tfrac{\Delta Z_{n+1}}{h}\big|^2 + 3(a_{11})^2 L_1' dh.
\end{aligned} \tag{57}$$

Keeping this in mind, we use arguments similar to those in (56) to obtain

$$\begin{aligned}
\big|\Phi_2^n - Y_n\big|^2 \le& 3(a_{21})^2 h^2\big|\nabla U(Y_n)\big|^2 + 3(a_{22})^2 h^2\big|\nabla U(\Phi_1^n)\big|^2 + 6(b_2)^2\big|\tfrac{\Delta Z_{n+1}}{h}\big|^2 \\
\le& 6(a_{21})^2 L_1^2 h^2\big|Y_n\big|^2 + 6(a_{22})^2 L_1^2 h^2\big|\Phi_1^n\big|^2 + 6(b_2)^2\big|\tfrac{\Delta Z_{n+1}}{h}\big|^2 + 3\big((a_{21})^2 + (a_{22})^2\big)L_1' dh \\
\le& \big(6(a_{21})^2 + 18(a_{22})^2 + 9(a_{11}a_{22})^2\big)L_1^2 h^2\big|Y_n\big|^2 + \big(6(b_2)^2 + 9(a_{22}b_1)^2\big)\big|\tfrac{\Delta Z_{n+1}}{h}\big|^2 \\
& + \big(3(a_{21})^2 + 3(a_{22})^2 + 6(a_{11}a_{22})^2\big)L_1' dh.
\end{aligned} \tag{58}$$

Combining (56) with (58) yields

$$I_1 \leq \kappa_1 L_1^2 h^2 |Y_n|^2 + \kappa_2 \left|\frac{\Delta Z_{n+1}}{h}\right|^2 + \kappa_3 L_1' dh, \tag{59}$$

where

$$\begin{aligned}
\kappa_1 :=&4\alpha^2(a_{11})^2 + 6\beta^2(a_{21})^2 + 18\beta^2(a_{22})^2 + 9\beta^2(a_{11}a_{22})^2,\\
\kappa_2 :=&4\alpha^2(b_1)^2 + 9\beta^2(a_{22}b_1)^2 + 6\beta^2(b_2)^2,\\
\kappa_3 :=&2\alpha^2(a_{11})^2 + 3\beta^2(a_{21})^2 + 3\beta^2(a_{22})^2 + 6\beta^2(a_{11}a_{22})^2.
\end{aligned} \tag{60}$$

Noting that $I_2 = \frac{2L_1}{\mu} I_1$, it follows immediately that

$$I_2 \leq \frac{2\kappa_1}{\mu} L_1^3 h^2 |Y_n|^2 + \frac{2\kappa_2}{\mu} L_1 \left|\frac{\Delta Z_{n+1}}{h}\right|^2 + \frac{2\kappa_3}{\mu} L_1 L_1' dh. \tag{61}$$

Under the stepsize condition $h \leq \frac{\mu}{4\kappa_1 L_1^2} \wedge \frac{\mu^2}{8\kappa_1 L_1^3}$, one can derive from (59) and (61) that

$$I_1 + I_2 \leq \tfrac{1}{2}\mu h |Y_n|^2 + \left(1 + \tfrac{2L_1}{\mu}\right)\kappa_2 \left|\frac{\Delta Z_{n+1}}{h}\right|^2 + \left(1 + \tfrac{2L_1}{\mu}\right)\kappa_3 L_1' dh. \tag{62}$$

Inserting this into (55) gives

$$\left|Y_{n+1}\right|^2 \leq \left(1 - \tfrac{3}{4}\mu h\right)|Y_n|^2 + 6|\Delta W_{n+1}|^2 + 2\sqrt{2}\langle Y_n, \Delta W_{n+1}\rangle + c_1 \left|\frac{\sqrt{3}\Delta Z_{n+1}}{h}\right|^2 + c_2 dh, \tag{63}$$

where

$$c_1 := \left(\tfrac{\sqrt{3}}{3} + \tfrac{2\sqrt{3}L_1}{3\mu}\right)\kappa_2, \quad c_2 := 2\mu' + 4L_1' + \kappa_3 L_1' + \tfrac{2\kappa_3}{\mu} L_1 L_1'. \tag{64}$$

Denote further

$$\Xi_{n+1} := 6|\Delta W_{n+1}|^2 + 2\sqrt{2}\langle Y_n, \Delta W_{n+1}\rangle + c_1 \left|\frac{\sqrt{3}\Delta Z_{n+1}}{h}\right|^2 + c_2 dh. \tag{65}$$

Taking the $p$-th power on both sides and then taking conditional expectation with respect to $\mathcal{F}_{t_n}$, together with the binomial expansion, yields

$$\mathbb{E}\left[|Y_{n+1}|^{2p}\Big|\mathcal{F}_{t_n}\right] \leq \mathbb{E}\left[\left(1 - \tfrac{3}{4}\mu h\right)|Y_n|^2 + \Xi_{n+1}\right)^p\Big|\mathcal{F}_{t_n}\right] =: \left(1 - \tfrac{3}{4}\mu h\right)^p |Y_n|^{2p} + R_{n+1}, \tag{66}$$

where for short we denote

$$R_{n+1} := \sum_{i=1}^{p} \mathcal{C}_i^p \left(1 - \tfrac{3}{4}\mu h\right)^{p-i} |Y_n|^{2p-2i} \mathbb{E}\left[\left(\Xi_{n+1}\right)^i\Big|\mathcal{F}_{t_n}\right], \quad \mathcal{C}_i^p := \frac{p!}{i!(p-i)!}. \tag{67}$$

Next, the conditional expectations $\mathbb{E}[(\Xi_{n+1})^i|\mathcal{F}_{t_n}]$ are treated separately for the cases $i = 1$ and $i \geq 2$. We first define

$$\Delta W_{n+1}^k := \int_{t_n}^{t_{n+1}} dW_t^k, \quad \Delta Z_{n+1}^k := \int_{t_n}^{t_{n+1}}\int_{t_n}^{t} dW_s^k \, dt, \quad k \in [d],$$

and apply the properties of the stochastic integral to get

$$\mathbb{E}\left[\left(\Delta W_{n+1}^k\right)^l\Big|\mathcal{F}_{t_n}\right] = \begin{cases} (l-1)!! h^{\frac{l}{2}} & \text{if } l \text{ is even,}\\ 0 & \text{if } l \text{ is odd,} \end{cases} \tag{68}$$

and

$$\mathbb{E}\left[\left(\frac{\sqrt{3}\Delta Z_{n+1}^k}{h}\right)^l\Big|\mathcal{F}_{t_n}\right] = \begin{cases} (l-1)!! \, h^{\frac{l}{2}} & \text{if } l \text{ is even,}\\ 0 & \text{if } l \text{ is odd,} \end{cases} \tag{69}$$

which implies

$$\mathbb{E}\left[\Delta W_{n+1}\Big|\mathcal{F}_{t_n}\right] = 0, \quad \mathbb{E}\left[|\Delta W_{n+1}|^2\Big|\mathcal{F}_{t_n}\right] = dh, \quad \mathbb{E}\left[\left|\frac{\sqrt{3}\Delta Z_{n+1}}{h}\right|^2\Big|\mathcal{F}_{t_n}\right] = dh. \tag{70}$$

Recalling the definition of $\Xi_{n+1}$, we have

$$\begin{aligned}
\mathbb{E}\left[\Xi_{n+1}\Big|\mathcal{F}_{t_n}\right] =&6\mathbb{E}\left[|\Delta W_{n+1}|^2\Big|\mathcal{F}_{t_n}\right] + 2\sqrt{2}\left\langle Y_n, \mathbb{E}\left[\Delta W_{n+1}\big|\mathcal{F}_{t_n}\right]\right\rangle + c_1 \mathbb{E}\left[\left|\frac{\sqrt{3}\Delta Z_{n+1}}{h}\right|^2\Big|\mathcal{F}_{t_n}\right] + c_2 dh\\
=&6dh + c_1 dh + c_2 dh.
\end{aligned} \tag{71}$$

Before coming to estimates for the case $i \geq 2$, we apply (54), (68) and (69) to get

$$\mathbb{E}\Big[\big|\Delta W_{n+1}\big|^{2i}\Big|\mathcal{F}_{t_n}\Big] \leq (2i-1)!!d^i h^i, \quad \mathbb{E}\Big[\big|\Delta W_{n+1}\big|^{i}\Big|\mathcal{F}_{t_n}\Big] \leq (i-1)!!d^{\frac{i}{2}}h^{\frac{i}{2}}, \quad \mathbb{E}\Big[\big|\tfrac{\sqrt{3}\Delta Z_{n+1}}{h}\big|^{2i}\Big|\mathcal{F}_{t_n}\Big] \leq (2i-1)!!d^i h^i. \tag{72}$$

Bearing this in mind and applying (54) yield

$$\begin{aligned}
\mathbb{E}\Big[\big(\Xi_{n+1}\big)^i\Big|\mathcal{F}_{t_n}\Big] \leq &4^{i-1}\Big(6^i\mathbb{E}\Big[\big|\Delta W_{n+1}\big|^{2i}\Big|\mathcal{F}_{t_n}\Big] + 2^{\frac{3i}{2}}\big|Y_n\big|^i\mathbb{E}\Big[\big|\Delta W_{n+1}\big|^{i}\Big|\mathcal{F}_{t_n}\Big] \\
&+ (c_1)^i\mathbb{E}\Big[\big|\tfrac{\sqrt{3}\Delta Z_{n+1}}{h}\big|^{2i}\Big|\mathcal{F}_{t_n}\Big] + (c_2)^i d^i h^i\Big) \\
\leq &\Big(4^{i-1}2^{\frac{3i}{2}}(i-1)!!d^{\frac{i}{2}}h^{\frac{i}{2}}\big|Y_n\big|^i + 4^{i-1}\big(6^i(2i-1)!! + (c_1)^i(2i-1)!! + (c_2)^i\big)d^i h^i\Big).
\end{aligned} \tag{73}$$

Inserting (71) and (73) into $R_{n+1}$, together with $h \leq 1 \wedge \frac{4}{\mu}$ and $(1-\frac{3}{4}\mu h)^k \leq (1-\frac{3}{4}\mu h) \leq 1$, $k \geq 1$, we obtain

$$\begin{aligned}
R_{n+1} \leq &\sum_{i=2}^{p}\mathcal{C}_i^p\big(1-\tfrac{3}{4}\mu h\big)^{p-i}4^{i-1}2^{\frac{3i}{2}}(i-1)!!d^{\frac{i}{2}}h^{\frac{i}{2}}\big|Y_n\big|^{2p-i} \\
&+ \sum_{i=1}^{p}\mathcal{C}_i^p\big(1-\tfrac{3}{4}\mu h\big)^{p-i}4^{i-1}\big(6^i(2i-1)!! + (c_1)^i(2i-1)!! + (c_2)^i\big)d^i h^i\big|Y_n\big|^{2p-2i} \\
\leq &h\sum_{i=2}^{p}\ell_1(i)d^{\frac{i}{2}}\big|Y_n\big|^{2p-i} + h\sum_{i=1}^{p}\ell_2(i)d^i\big|Y_n\big|^{2p-2i},
\end{aligned} \tag{74}$$

where

$$\ell_1(i) := \mathcal{C}_i^p 4^{i-1}2^{\frac{3i}{2}}(i-1)!!, \quad \ell_2(i) := \mathcal{C}_i^p 4^{i-1}\big(6^i(2i-1)!! + (c_1)^i(2i-1)!! + (c_2)^i\big). \tag{75}$$

The Young inequality with $\varepsilon_1, \varepsilon_2 > 0$ implies

$$d^{\frac{i}{2}}\big|Y_n\big|^{2p-i} \leq \varepsilon_1\big|Y_n\big|^{2p} + C_{i,p}(\varepsilon_1)\big(\ell_1(i)\big)^{\frac{2p}{i}}d^p, \quad d^i\big|Y_n\big|^{2p-2i} \leq \varepsilon_2\big|Y_n\big|^{2p} + C_{i,p}(\varepsilon_2)\big(\ell_2(i)\big)^{\frac{p}{i}}d^p, \tag{76}$$

where $C_{i,p}(\varepsilon_1) = \frac{i}{2p}\big(\frac{2p-i}{2p\varepsilon_1}\big)^{\frac{2p-i}{i}}$ and $C_{i,p}(\varepsilon_2) = \frac{i}{p}\big(\frac{p-i}{p\varepsilon_2}\big)^{\frac{p-i}{i}}$. Thus, we have

$$R_{n+1} \leq \big((p-1)\varepsilon_1 h + p\varepsilon_1 h\big)\big|Y_n\big|^{2p} + \Big(\sum_{i=2}^{p}C_{i,p}(\varepsilon_1)\big(\ell_1(i)\big)^{\frac{2p}{i}} + \sum_{i=1}^{p}C_{i,p}(\varepsilon_2)\big(\ell_1(i)\big)^{\frac{2p}{i}}\Big)d^p h. \tag{77}$$

Plugging this into (66) and noting $(1-\frac{3}{4}\mu h)^k \leq (1-\frac{3}{4}\mu h) \leq 1$, $k \geq 1$, one can choose $\varepsilon_1 = \frac{\mu}{4(p-1)}$ and $\varepsilon_2 = \frac{\mu}{4p}$ to get

$$\mathbb{E}\Big[\big|Y_{n+1}\big|^{2p}\Big|\mathcal{F}_{t_n}\Big] \leq \big(1-\tfrac{\mu}{4}h\big)\big|Y_n\big|^{2p} + \Big(\sum_{i=2}^{p}C_{i,p}\big(\tfrac{\mu}{4(p-1)}\big)\big(\ell_1(i)\big)^{\frac{2p}{i}} + \sum_{i=1}^{p}C_{i,p}\big(\tfrac{\mu}{4p}\big)\big(\ell_2(i)\big)^{\frac{2p}{i}}\Big)d^p h, \tag{78}$$

further indicating

$$\begin{aligned}
\mathbb{E}\Big[\big|Y_{n+1}\big|^{2p}\Big] \leq &\big(1-\tfrac{\mu}{4}h\big)\mathbb{E}\Big[\big|Y_n\big|^{2p}\Big] + \Big(\sum_{i=2}^{p}C_{i,p}\big(\tfrac{\mu}{4(p-1)}\big)\big(\ell_1(i)\big)^{\frac{2p}{i}} + \sum_{i=1}^{p}C_{i,p}\big(\tfrac{\mu}{4p}\big)\big(\ell_2(i)\big)^{\frac{2p}{i}}\Big)d^p h \\
\leq &\big(1-\tfrac{\mu}{4}h\big)^{n+1}\mathbb{E}\Big[\big|X_0\big|^{2p}\Big] + \tfrac{4}{\mu}\Big(\sum_{i=2}^{p}C_{i,p}\big(\tfrac{\mu}{4(p-1)}\big)\big(\ell_1(i)\big)^{\frac{2p}{i}} + \sum_{i=1}^{p}C_{i,p}\big(\tfrac{\mu}{4p}\big)\big(\ell_2(i)\big)^{\frac{2p}{i}}\Big)d^p \\
\leq &e^{-\frac{\mu}{4}t_{n+1}}\mathbb{E}\Big[\big|X_0\big|^{2p}\Big] + \mathcal{M}_2(p)d^p,
\end{aligned} \tag{79}$$

where we have used the fact that for any $x > 0$, $1 - x \leq e^{-x}$ and denote

$$\mathcal{M}_2(p) := \tfrac{4}{\mu}\Big(\sum_{i=2}^{p}C_{i,p}\big(\tfrac{\mu}{4(p-1)}\big)\big(\ell_1(i)\big)^{\frac{2p}{i}} + \sum_{i=1}^{p}C_{i,p}\big(\tfrac{\mu}{4p}\big)\big(\ell_2(i)\big)^{\frac{2p}{i}}\Big). \tag{80}$$

This completes the proof. $\qquad\square$

# B. Proof of Proposition 3.3

This section is devoted to the proof of Proposition 3.3, which characterizes the finite-time convergence of the accelerated RKLMC (10) algorithm. The proof relies on the strong fundamental convergence theorem for SDEs (Milstein & Tretyakov, 2004; Yang & Wang, 2025). Accordingly, we first carry out a detailed analysis of the one-step errors between the exact Langevin dynamics and its accelerated numerical approximation.

## B.1. One-Step Approximations

In this subsection, we introduce the one-step representations of the exact Langevin dynamics, the TELMC algorithm and the RKLMC algorithm, which serve as the basis for the subsequent local error analysis. First, let $X(s, x; t)$ denote the solution of Langevin SDE (2) at time $t$, starting from the initial value $x \in \mathbb{R}^d$ at time $s$, which is given by

$$X(s, x; t) = x - \int_s^t \nabla U(X_r) \, dr + \int_s^t \sqrt{2} \, dW_r. \tag{81}$$

Then the exact one-step solution of the Langevin SDE (2) admits the following representation:

$$X(t, x; t+h) = x - \int_t^{t+h} \nabla U(X_s) ds + \sqrt{2} \Delta W_{t,t+h}, \quad \forall x \in \mathbb{R}^d, \quad t \geq 0, \tag{82}$$

By applying the Itô formula to $\nabla U(X_s)$, one can easily obtain

$$\begin{aligned} X(t, x; t+h) = & x - \nabla U(x)h + \sqrt{2} \Delta W_{t,t+h} + \int_t^{t+h} \int_t^s \nabla^2 U(X_r) \nabla U(X_r) dr ds \\ & - \int_t^{t+h} \int_t^s \nabla(\Delta U(X_r)) dr ds - \sqrt{2} \int_t^{t+h} \int_t^s \nabla^2 U(X_r) dW_r ds. \end{aligned} \tag{83}$$

Likewise, let $\mathbb{Y}(t, x; t+h)$ denote the one-step approximation at time $t + h$ generated by the TELMC scheme (7), starting from $x \in \mathbb{R}^d$ at time $t$, defined by

$$\mathbb{Y}(t, x; t+h) := x - \nabla U(x)h + \sqrt{2} \Delta W_{t,t+h} + \tfrac{1}{2} \nabla^2 U(x) \nabla U(x) h^2 - \tfrac{1}{2} \nabla(\Delta U(x)) h^2 - \sqrt{2} \nabla^2 U(x) \Delta Z_{t,t+h}. \tag{84}$$

Here, $\Delta W_{t,t+h}$ and $\Delta Z_{t,t+h}$ are defined componentwise by

$$\begin{aligned} \Delta W_{t,t+h} &:= \left(\Delta W_{t,t+h}^1, \Delta W_{t,t+h}^2, \cdots, \Delta W_{t,t+h}^d\right)^T, & \Delta W_{t,t+h}^k &:= \int_t^{t+h} dW_s^k, \quad k \in [d], \\ \Delta Z_{t,t+h} &:= \left(\Delta Z_{t,t+h}^1, \Delta Z_{t,t+h}^2, \cdots, \Delta Z_{t,t+h}^d\right)^T, & \Delta Z_{t,t+h}^k &:= \int_t^{t+h} \int_t^s dW_r^k \, ds. \end{aligned} \tag{85}$$

Finally, the one-step approximation scheme associated with the RKLMC algorithm (10) is defined by

$$Y(t, x; t+h) := x - (1 - \alpha - \beta) \nabla U(x)h - \alpha \nabla U(\Phi_1^h)h - \beta \nabla U(\Phi_2^h)h + \sqrt{2} \Delta W_{t,t+h}, \tag{86}$$

where the stages $\Phi_1^h$ and $\Phi_2^h$ are given by

$$\begin{aligned} \Phi_1^h &:= x - a_{11} \nabla U(x)h + \sqrt{2} b_1 \tfrac{\Delta Z_{t,t+h}}{h}, \\ \Phi_2^h &:= x - a_{21} \nabla U(x)h - a_{22} \nabla U(\Phi_1^h)h + \sqrt{2} b_2 \tfrac{\Delta Z_{t,t+h}}{h}. \end{aligned} \tag{87}$$

## B.2. One-Step Strong and Weak Errors

This subsection is devoted to the analysis of one-step strong and weak errors between the exact solution of the Langevin SDE (83) and the RKLMC scheme (86). To facilitate the analysis, we decompose the one-step errors between the Langevin dynamics and RKLMC into two components:

$$\underbrace{X(t, x; t+h) - Y(t, x; t+h)}_{\text{One-step errors between Langevin and RKLMC}} = \underbrace{X(t, x; t+h) - \mathbb{Y}(t, x; t+h)}_{\text{One-step errors between Langevin and TELMC}} + \underbrace{\mathbb{Y}(t, x; t+h) - Y(t, x; t+h)}_{\text{One-step errors between TELMC and RKLMC}}. \tag{88}$$

B.2.1. ONE-STEP STRONG AND WEAK ERRORS BETWEEN THE LANGEVIN DYNAMICS AND TELMC

In this subsection we focus on the discrepancy between the exact one-step solution of the Langevin SDE (83) and the TELMC scheme (84). At first, we establish an estimate of the Hölder continuity in time for the Langevin SDE (2). The proof of this result can be found in Lemma A.1 of (Wang & Yang, 2026).

**Lemma B.1** (Hölder continuity in time for Langevin dynamics). *Let Assumptions 2.1, 2.2 hold and let $X(s, x; t)$ denote the solution of Langevin SDE (2) at time $t$, starting from the initial value $x \in \mathbb{R}^d$ at time $s$. If the uniform stepsize satisfies $h \leq 1 \wedge \frac{1}{2L_1} \wedge \frac{1}{2L_1'}$, then for any $0 < \theta \leq h$ and any $p \geq 1$, it holds*

$$\mathbb{E}\Big[\big|X(s, x; t + \theta) - X(s, x; t)\big|^{2p}\Big] \leq \big(\mathcal{H}_1(p)d^p + \mathcal{H}_2(p)|x|^{2p}\big)\theta^p, \tag{89}$$

*where*

$$\mathcal{H}_1(p) := 2^{3p-2}L_1'^p + 2^{3p-2}\mathcal{M}_1(p)L_1^p + 2^{3p-1}(2p-1)!!, \quad \mathcal{H}_2(p) := 2^{3p-2}L_1^p. \tag{90}$$

We are now ready to quantify the one-step strong and weak errors between the Langevin dynamics and the TELMC scheme, summarized in the following lemma.

**Lemma B.2.** *Let Assumptions 2.1-2.4 hold and let the timestep $h$ satisfy $h \leq 1 \wedge \frac{1}{2L_1'} \wedge \frac{1}{2L_1}$. Then for any $x \in \mathbb{R}^d$ and any $t \geq 0$, it holds*

$$\begin{aligned}
\Big|\mathbb{E}\Big[X(t, x; t + h) - \mathbb{Y}(t, x; t + h)\Big]\Big| &\leq \Big(K_{1,1}^E d^3 + K_{1,2}^E |x|^6\Big)^{\frac{1}{2}} h^{\frac{5}{2}}, \\
\Big(\mathbb{E}\Big[\big|X(t, x; t + h) - \mathbb{Y}(t, x; t + h)\big|^2\Big]\Big)^{1/2} &\leq \Big(K_{1,1}^M d^3 + K_{1,2}^M |x|^6\Big)^{1/2} h^2,
\end{aligned} \tag{91}$$

*where*

$$\begin{aligned}
K_{1,1}^E :=&\, \mathcal{H}_1(1)L_1^4 + 2\mathcal{H}_1(1)L_1'^2 L_2^2 + 2\mathcal{H}_1(1)L_1^2 L_2^2 + \mathcal{H}_2(1)L_1^4 + 2\mathcal{H}_2(1)L_1'^2 L_2^2 + 2\mathcal{H}_2(1)L_1^2 L_2^2 + \mathcal{H}_1(1)L_3^2 + \mathcal{H}_2(1)L_3^2, \\
K_{1,2}^E :=&\, \mathcal{H}_1(1)L_1^2 L_2^2 + \mathcal{H}_2(1)L_1'^2 L_2^2 + \mathcal{H}_2(1)L_1^4 + 2\mathcal{H}_2(1)L_1^2 L_2^2 + \mathcal{H}_2(1)L_3^2, \\
K_{1,1}^M :=&\, 3\Big(\mathcal{H}_1(1)L_1^4 + 2\mathcal{H}_1(1)L_1'^2 L_2^2 + 2\mathcal{H}_1(1)L_1^2 L_2^2 + \mathcal{H}_2(1)L_1^4 + 2\mathcal{H}_2(1)L_1'^2 L_2^2 + 2\mathcal{H}_2(1)L_1^2 L_2^2 + \mathcal{H}_1(1)L_3^2 \\
&\, + \mathcal{H}_2(1)L_3^2 + \mathcal{H}_1(1)L_2^2 + \mathcal{H}_2(1)L_2^2\Big), \\
K_{1,2}^M :=&\, 3\Big(\mathcal{H}_1(1)L_1^2 L_2^2 + \mathcal{H}_2(1)L_1'^2 L_2^2 + \mathcal{H}_2(1)L_1^4 + 2\mathcal{H}_2(1)L_1^2 L_2^2 + \mathcal{H}_2(1)L_3^2 + \mathcal{H}_2(1)L_2^2\Big),
\end{aligned} \tag{92}$$

*and constants $\mathcal{H}_i(1)$, $i = 1, 2$ come from (90).*

*Proof.* Subtracting (84) from (83) results in

$$\begin{aligned}
X(t, x; t + h) - \mathbb{Y}(t, x; t + h) =&\, \underbrace{\int_t^{t+h} \int_t^s \big(\nabla^2 U(X_r)\nabla U(X_r) - \nabla^2 U(x)\nabla U(x)\big)\mathrm{d}r\mathrm{d}s}_{=:R_{t+h}^1} \\
&\, \underbrace{-\int_t^{t+h} \int_t^s \big(\nabla(\Delta U(X_r)) - \nabla(\Delta U(x))\big)\mathrm{d}r\mathrm{d}s}_{=:R_{t+h}^2} \\
&\, \underbrace{-\sqrt{2}\int_t^{t+h} \int_t^s \big(\nabla^2 U(X_r) - \nabla^2 U(x)\big)\mathrm{d}W_r\mathrm{d}s}_{=:R_{t+h}^3}.
\end{aligned} \tag{93}$$

Applying the triangle inequality yields

$$\Big|\mathbb{E}\Big[X(t, x; t + h) - \mathbb{Y}(t, x; t + h)\Big]\Big| \leq \sum_{i=1}^3 \Big|\mathbb{E}\big[R_{t+h}^i\big]\Big|. \tag{94}$$

Thanks to (54), we also have

$$\mathbb{E}\Big[\big|X(t,x;t+h) - \mathbb{Y}(t,x;t+h)\big|^2\Big] \le 3\sum_{i=1}^{3}\mathbb{E}\Big[\big|R_{t+h}^i\big|^2\Big]. \tag{95}$$

In what follows, we estimate expectations and second moments of terms $R_{t+h}^i$, $i = 1, 2, 3$. The overall strategy is to first bound the second moments of each term and then derive the corresponding bounds on their expectations.

**Estimate of $R_{t+h}^1$:** We decompose $R_{t+h}^1$ as

$$R_{t+h}^1 = \underbrace{\int_t^{t+h}\int_t^s \nabla^2 U(X_r)\big(\nabla U(X_r) - \nabla U(x)\big)\mathrm{d}r\,\mathrm{d}s}_{=:R_{t+h}^{1,1}} + \underbrace{\int_t^{t+h}\int_t^s \big(\nabla^2 U(X_r) - \nabla^2 U(x)\big)\nabla U(x)\mathrm{d}r\,\mathrm{d}s}_{=:R_{t+h}^{1,2}}. \tag{96}$$

In what follows, the two terms $R_{t+h}^{1,1}$ and $R_{t+h}^{1,2}$ are estimated separately. By invoking Lemma B.1 together with the Hölder inequality, the gradient Lipschitz condition (22) and (43), we obtain

$$\begin{aligned}
\mathbb{E}\Big[\big|R_{t+h}^{1,1}\big|^2\Big] &\le h^2\int_t^{t+h}\int_t^s \mathbb{E}\Big[\big|\nabla^2 U(X_r)\big(\nabla U(X_r) - \nabla U(x)\big)\big|^2\Big]\mathrm{d}r\,\mathrm{d}s \\
&\le L_1^2 h^2\int_t^{t+h}\int_t^s \mathbb{E}\Big[\big|\nabla U(X_r) - \nabla U(x)\big|^2\Big]\mathrm{d}r\,\mathrm{d}s \\
&\le L_1^4 h^2\int_t^{t+h}\int_t^s \mathbb{E}\big[|X_r - x|^2\big]\mathrm{d}r\,\mathrm{d}s \\
&\le \Big(\mathcal{H}_1(1)L_1^4 d + \mathcal{H}_2(1)L_1^4|x|^2\Big)h^5 \\
&\le \Big(\big(\mathcal{H}_1(1)L_1^4 + \mathcal{H}_2(1)L_1^4\big)d^3 + \mathcal{H}_2(1)L_1^4|x|^6\Big)h^5,
\end{aligned} \tag{97}$$

where in the last step we used the elementary inequality

$$d^i \le d^j, \quad |x|^{2i} \le d^j + |x|^{2j}, \quad i \le j. \tag{98}$$

Analogously, using the linear growth condition (23) and the Hessian Lipschitz condition (24) additionally shows

$$\begin{aligned}
\mathbb{E}\Big[\big|R_{t+h}^{1,2}\big|^2\Big] &\le h^2\int_t^{t+h}\int_t^s \mathbb{E}\Big[\big|\big(\nabla^2 U(X_r) - \nabla^2 U(x)\big)\nabla U(x)\big|^2\Big]\mathrm{d}r\,\mathrm{d}s \\
&\le 2L_2^2\big(L_1'^2 d + L_1^2|x|^2\big)h^2\int_t^{t+h}\int_t^s \mathbb{E}\big[|X_r - x|^2\big]\mathrm{d}r\,\mathrm{d}s \\
&\le 2L_2^2\big(L_1'^2 d + L_1^2|x|^2\big)\big(\mathcal{H}_1(1)d + \mathcal{H}_2(1)|x|^2\big)h^5 \\
&\le \Big(\big(2\mathcal{H}_1(1)L_1'^2 L_2^2 + \mathcal{H}_1(1)L_1^2 L_2^2 + \mathcal{H}_2(1)L_1'^2 L_2^2\big)d^2 \\
&\quad + \big(\mathcal{H}_1(1)L_1^2 L_2^2 + \mathcal{H}_2(1)L_1'^2 L_2^2 + 2\mathcal{H}_2(1)L_1^2 L_2^2\big)|x|^4\Big)h^5 \\
&\le \Big(\big(2\mathcal{H}_1(1)L_1'^2 L_2^2 + 2\mathcal{H}_1(1)L_1^2 L_2^2 + 2\mathcal{H}_2(1)L_1'^2 L_2^2 + 2\mathcal{H}_2(1)L_1^2 L_2^2\big)d^3 \\
&\quad + \big(\mathcal{H}_1(1)L_1^2 L_2^2 + \mathcal{H}_2(1)L_1'^2 L_2^2 + 2\mathcal{H}_2(1)L_1^2 L_2^2\big)|x|^6\Big)h^5.
\end{aligned} \tag{99}$$

Combining (97) with (99) and using the inequality (54), we conclude that

$$\mathbb{E}\Big[\big|R_{t+h}^1\big|^2\Big] \le \Big(\mathcal{M}_1^{R,1}d^3 + \mathcal{M}_2^{R,1}|x|^6\Big)h^5, \tag{100}$$

where

$$\begin{aligned}
\mathcal{M}_1^{R,1} &:= \mathcal{H}_1(1)L_1^4 + 2\mathcal{H}_1(1)L_1'^2 L_2^2 + 2\mathcal{H}_1(1)L_1^2 L_2^2 + \mathcal{H}_2(1)L_1^4 + 2\mathcal{H}_2(1)L_1'^2 L_2^2 + 2\mathcal{H}_2(1)L_1^2 L_2^2, \\
\mathcal{M}_2^{R,1} &:= \mathcal{H}_1(1)L_1^2 L_2^2 + \mathcal{H}_2(1)L_1'^2 L_2^2 + \mathcal{H}_2(1)L_1^4 + 2\mathcal{H}_2(1)L_1^2 L_2^2.
\end{aligned} \tag{101}$$

Keeping this in mind and using the Hölder inequality, we obtain

$$\left|\mathbb{E}\big[R^1_{t+h}\big]\right| \leq \left(\mathbb{E}\big[\big|R^1_{t+h}\big|^2\big]\right)^{\frac{1}{2}} \leq \left(\mathcal{M}^{R,1}_1 d^3 + \mathcal{M}^{R,1}_2 |x|^6\right)^{\frac{1}{2}} h^{\frac{5}{2}}. \tag{102}$$

**Estimate of $R^2_{t+h}$:** Using the Hölder inequality together with Lemma B.1, one can apply (50) and (98) to get

$$\begin{aligned}
\mathbb{E}\big[\big|R^2_{t+h}\big|^2\big] &\leq h^2 \int_t^{t+h} \int_t^s \mathbb{E}\big[\big|\nabla(\Delta U(X_r)) - \nabla(\Delta U(x))\big|^2\big] \mathrm{d}r\,\mathrm{d}s \\
&\leq L_3^2 d^2 h^2 \int_t^{t+h} \int_t^s \mathbb{E}\big[|X_r - x|^2\big] \mathrm{d}r\,\mathrm{d}s \\
&\leq \left(\mathcal{H}_1(1) L_3^2 d^3 + \mathcal{H}_2(1) L_3^2 d^2 |x|^2\right) h^5 \\
&\leq \left(\mathcal{M}^{R,2}_1 d^3 + \mathcal{M}^{R,2}_2 |x|^6\right) h^5,
\end{aligned} \tag{103}$$

where

$$\mathcal{M}^{R,2}_1 =: \mathcal{H}_1(1) L_3^2 + \mathcal{H}_2(1) L_3^2, \quad \mathcal{M}^{R,2}_2 =: \mathcal{H}_2(1) L_3^2. \tag{104}$$

Using the same arguments as estimating $|\mathbb{E}[R^1_{t+h}]|$, we immediately arrive at

$$\left|\mathbb{E}\big[R^2_{t+h}\big]\right| \leq \left(\mathcal{M}^{R,2}_1 d^3 + \mathcal{M}^{R,2}_2 |x|^6\right)^{\frac{1}{2}} h^{\frac{5}{2}}. \tag{105}$$

**Estimate of $R^3_{t+h}$:** Applying (98), the Hessian Lipschitz condition (24), the Hölder inequality and the Itô isometry shows

$$\begin{aligned}
\mathbb{E}\big[\big|R^3_{t+h}\big|^2\big] &\leq h \int_t^{t+h} \mathbb{E}\left[\left|\int_t^s \nabla^2 U(X_r) - \nabla^2 U(x)\mathrm{d}W_r\right|^2\right] \mathrm{d}s \\
&= h \int_t^{t+h} \int_t^s \mathbb{E}\big[\big|\nabla^2 U(X_r) - \nabla^2 U(x)\big|^2\big] \mathrm{d}r\,\mathrm{d}s \\
&\leq L_2^2 h \int_t^{t+h} \int_t^s \mathbb{E}\big[|X_r - x|^2\big] \mathrm{d}r\,\mathrm{d}s \\
&\leq \left(\mathcal{H}_1(1) L_2^2 d + \mathcal{H}_2(1) L_2^2 |x|^2\right) h^4 \\
&\leq \left(\mathcal{M}^{R,3}_1 d^3 + \mathcal{M}^{R,3}_2 |x|^6\right) h^4,
\end{aligned} \tag{106}$$

where

$$\mathcal{M}^{R,3}_1 =: \mathcal{H}_1(1) L_2^2 + \mathcal{H}_2(1) L_2^2, \quad \mathcal{M}^{R,3}_2 =: \mathcal{H}_2(1) L_2^2. \tag{107}$$

Thanks to the basic property of the stochastic integral, one can easily see

$$\left|\mathbb{E}\big[R^3_{t+h}\big]\right| = 0. \tag{108}$$

Combining the above estimates, we obtain the desired result from (94) and (95). □

### B.2.2. ONE-STEP STRONG AND WEAK ERRORS BETWEEN THE TELMC AND RKLMC

Following the same strategy as in the previous subsubsection, we analyze the one-step discrepancy between the TELMC scheme (84) and the RKLMC scheme (86). To this end, we first perform a Taylor expansion of $\nabla U(\Phi^h_i)$, $i = 1, 2$, around the point $x$, to show

$$\begin{aligned}
\nabla U(\Phi^h_i) =& \nabla U(x) + \nabla^2 U(x)(\Phi^h_i - x) + \tfrac{1}{2}\nabla^3 U(x)(\Phi^h_i - x, \Phi^h_i - x) \\
&+ \tfrac{1}{2}\int_0^1 (1-t)^2 \nabla^4 U(x + t(\Phi^h_i - x))\big(\Phi^h_i - x, \Phi^h_i - x, \Phi^h_i - x\big) \mathrm{d}t.
\end{aligned} \tag{109}$$

Recalling the one-step stages defined by (87), we have

$$\Phi_1^h - x := -a_{11}\nabla U(x)h + \sqrt{2}b_1\frac{\Delta Z_{t,t+h}}{h},$$
$$\Phi_2^h - x := -a_{21}\nabla U(x)h - a_{22}\nabla U(\Phi_1^h)h + \sqrt{2}b_2\frac{\Delta Z_{t,t+h}}{h}, \tag{110}$$

which in turn implies

$$\nabla^2 U(x)(\Phi_1^h - x) = -a_{11}\nabla^2 U(x)\nabla U(x)h + \sqrt{2}b_1\nabla^2 U(x)\frac{\Delta Z_{t,t+h}}{h},$$
$$\nabla^2 U(x)(\Phi_2^h - x) = -a_{21}\nabla^2 U(x)\nabla U(x)h - a_{22}\nabla^2 U(x)\nabla U(\Phi_1^h)h + \sqrt{2}b_2\nabla^2 U(x)\frac{\Delta Z_{t,t+h}}{h}. \tag{111}$$

Moreover, since $U \in \mathcal{C}^4(\mathbb{R}^d, \mathbb{R})$, the third-order derivative $\nabla^3 U(x)$ is symmetric. In particular, for any $x, v_1, v_2 \in \mathbb{R}^d$, the bilinearity and symmetry of $\nabla^3 U(x)$ imply

$$\nabla^3 U(x)(v_1 + v_2, v_1 + v_2) = \nabla^3 U(x)(v_1, v_1) + \nabla^3 U(x)(v_1, v_2) + \nabla^3 U(x)(v_2, v_1) + \nabla^3 U(x)(v_2, v_2)$$
$$= \nabla^3 U(x)(v_1, v_1) + 2\nabla^3 U(x)(v_1, v_2) + \nabla^3 U(x)(v_2, v_2).$$

Employing the above identity with $v_1$ and $v_2$ chosen according to the expressions of $\Phi_i^h - x$, $i = 1, 2$, we obtain

$$\nabla^3 U(x)(\Phi_1^h - x, \Phi_1^h - x) = (a_{11})^2 h^2 \nabla^3 U(x)\big(\nabla U(x), \nabla U(x)\big) - 2\sqrt{2}a_{11}b_1\nabla^3 U(x)\big(\nabla U(x), \Delta Z_{t,t+h}\big)$$
$$+ 2(b_1)^2\nabla^3 U(x)\big(\tfrac{\Delta Z_{t,t+h}}{h}, \tfrac{\Delta Z_{t,t+h}}{h}\big),$$
$$\nabla^3 U(x)(\Phi_2^h - x, \Phi_2^h - x) = h^2\nabla^3 U(x)\big(a_{21}\nabla U(x) + a_{22}\nabla U(\Phi_1^h), a_{21}\nabla U(x) + a_{22}\nabla U(\Phi_1^h)\big) \tag{112}$$
$$- 2\sqrt{2}b_2\nabla^3 U(x)\big(a_{21}\nabla U(x) + a_{22}\nabla U(\Phi_1^h), \Delta Z_{t,t+h}\big)$$
$$+ 2(b_2)^2\nabla^3 U(x)\big(\tfrac{\Delta Z_{t,t+h}}{h}, \tfrac{\Delta Z_{t,t+h}}{h}\big).$$

Substituting (111) and (112) into (109) yields

$$\nabla U(\Phi_1^h) = \nabla U(x) - a_{11}\nabla^2 U(x)\nabla U(x)h + \sqrt{2}b_1\nabla^2 U(x)\frac{\Delta Z_{t,t+h}}{h} + (b_1)^2\nabla^3 U(x)\big(\tfrac{\Delta Z_{t,t+h}}{h}, \tfrac{\Delta Z_{t,t+h}}{h}\big)$$
$$+ \tfrac{1}{2}(a_{11})^2 h^2 \nabla^3 U(x)\big(\nabla U(x), \nabla U(x)\big) - \sqrt{2}a_{11}b_1\nabla^3 U(x)\big(\nabla U(x), \Delta Z_{t,t+h}\big)$$
$$+ \tfrac{1}{2}\int_0^1 (1-t)^2\nabla^4 U(x + t(\Phi_1^h - x))(\Phi_1^h - x, \Phi_1^h - x, \Phi_1^h - x)\, \mathrm{d}t.$$
$$\nabla U(\Phi_2^h) = \nabla U(x) - a_{21}\nabla^2 U(x)\nabla U(x)h - a_{22}\nabla^2 U(x)\nabla U(\Phi_1^h)h + \sqrt{2}b_2\nabla^2 U(x)\frac{\Delta Z_{t,t+h}}{h} \tag{113}$$
$$+ (b_2)^2\nabla^3 U(x)\big(\tfrac{\Delta Z_{t,t+h}}{h}, \tfrac{\Delta Z_{t,t+h}}{h}\big) - \sqrt{2}b_2\nabla^3 U(x)\big(a_{21}\nabla U(x) + a_{22}\nabla U(\Phi_1^h), \Delta Z_{t,t+h}\big)$$
$$+ \tfrac{h^2}{2}\nabla^3 U(x)\big(a_{21}\nabla U(x) + a_{22}\nabla U(\Phi_1^h), a_{21}\nabla U(x) + a_{22}\nabla U(\Phi_1^h)\big)$$
$$+ \tfrac{1}{2}\int_0^1 (1-t)^2\nabla^4 U(x + t(\Phi_2^h - x))(\Phi_2^h - x, \Phi_2^h - x, \Phi_2^h - x)\, \mathrm{d}t.$$

Plugging the above expansions into (86), we rewrite the one-step RKLMC approximation $Y(t, x; t + h)$ in the explicit form:

$$Y(t, x; t+h) = x - \nabla U(x)h + \sqrt{2}\Delta W_{t,t+h} + \big(\alpha a_{11} + \beta a_{21}\big)h^2\nabla^2 U(x)\nabla U(x) + \beta a_{22}h^2\nabla^2 U(x)\nabla U(\Phi_1^h)$$
$$- \sqrt{2}\big(\alpha b_1 + \beta b_2\big)\nabla^2 U(x)\Delta Z_{t,t+h} - \big(\alpha(b_1)^2 + \beta(b_2)^2\big)h\nabla^3 U(x)\big(\tfrac{\Delta Z_{t,t+h}}{h}, \tfrac{\Delta Z_{t,t+h}}{h}\big)$$
$$- \tfrac{\alpha}{2}(a_{11})^2 h^3\nabla^3 U(x)\big(\nabla U(x), \nabla U(x)\big) + \sqrt{2}\alpha a_{11}b_1 h\nabla^3 U(x)\big(\nabla U(x), \Delta Z_{t,t+h}\big)$$
$$- \tfrac{\beta}{2}h^3\nabla^3 U(x)\big(a_{21}\nabla U(x) + a_{22}\nabla U(\Phi_1^h), a_{21}\nabla U(x) + a_{22}\nabla U(\Phi_1^h)\big)$$
$$+ \sqrt{2}\beta b_2 h\nabla^3 U(x)\big(a_{21}\nabla U(x) + a_{22}\nabla U(\Phi_1^h), \Delta Z_{t,t+h}\big) \tag{114}$$
$$- \tfrac{\alpha}{2}h\int_0^1 (1-t)^2\nabla^4 U(x + t(\Phi_1^h - x))(\Phi_1^h - x, \Phi_1^h - x, \Phi_1^h - x)\, \mathrm{d}t$$
$$- \tfrac{\beta}{2}h\int_0^1 (1-t)^2\nabla^4 U(x + t(\Phi_2^h - x))(\Phi_2^h - x, \Phi_2^h - x, \Phi_2^h - x)\, \mathrm{d}t.$$

Recalling the order conditions (12)-(14) and noting that

$$\nabla^2 U(x)\nabla U(\Phi_1^h) = \nabla^2 U(x)\nabla U(x) + \nabla^2 U(x)\big(\nabla U(\Phi_1^h) - \nabla U(x)\big),$$

we obtain

$$
\begin{aligned}
Y(t,x;t+h) =& x - \nabla U(x)h + \tfrac{1}{2}\nabla^2 U(x)\nabla U(x)h^2 - \sqrt{2}\nabla^2 U(x)\Delta Z_{t,t+h} - \tfrac{3}{2h}\nabla^3 U(x)\big(\Delta Z_{t,t+h}, \Delta Z_{t,t+h}\big) \\
& + \sqrt{2}\Delta W_{t,t+h} + \beta a_{22}h^2\nabla^2 U(x)\big(\nabla U(\Phi_1^h) - \nabla U(x)\big) - \tfrac{\alpha}{2}(a_{11})^2 h^3\nabla^3 U(x)\big(\nabla U(x), \nabla U(x)\big) \\
& - \tfrac{\beta}{2}h^3\nabla^3 U(x)\big(a_{21}\nabla U(x) + a_{22}\nabla U(\Phi_1^h), a_{21}\nabla U(x) + a_{22}\nabla U(\Phi_1^h)\big) \\
& + \sqrt{2}h\nabla^3 U(x)\Big(\big(\alpha a_{11}b_1 + \beta a_{21}b_2\big)\nabla U(x) + \beta a_{22}b_2\nabla U(\Phi_1^h), \Delta Z_{t,t+h}\Big) \\
& - \tfrac{\alpha}{2}h\int_0^1 (1-t)^2\nabla^4 U(x + t(\Phi_1^h - x))\,\mathrm{d}t\,(\Phi_1^h - x, \Phi_1^h - x, \Phi_1^h - x) \\
& - \tfrac{\beta}{2}h\int_0^1 (1-t)^2\nabla^4 U(x + t(\Phi_2^h - x))\,\mathrm{d}t\,(\Phi_2^h - x, \Phi_2^h - x, \Phi_2^h - x).
\end{aligned}
\tag{115}
$$

Subtracting (86) from (84), we decompose the one-step discrepancy into a sum of remainder terms as follows:

$$
\begin{aligned}
& \mathbb{Y}(t,x;t+h) - Y(t,x;t+h) \\
=& \underbrace{\tfrac{3}{2h}\nabla^3 U(x)\big(\Delta Z_{t,t+h}, \Delta Z_{t,t+h}\big) - \tfrac{1}{2}h^2\nabla(\Delta U(x))}_{=:R_{t+h}^4} \underbrace{-\beta a_{22}h^2\nabla^2 U(x)\big(\nabla U(\Phi_1^h) - \nabla U(x)\big)}_{=:R_{t+h}^5} \\
& + \underbrace{\big(\tfrac{\alpha}{2}(a_{11})^2 + \tfrac{\beta}{2}(a_{21})^2\big)h^3\nabla^3 U(x)\big(\nabla U(x), \nabla U(x)\big)}_{=:R_{t+h}^6} + \underbrace{\beta a_{21}a_{22}h^3\nabla^3 U(x)\big(\nabla U(x), \nabla U(\Phi_1^h)\big)}_{=:R_{t+h}^7} \\
& + \underbrace{\tfrac{\beta}{2}(a_{22})^2 h^3\nabla^3 U(x)\big(\nabla U(\Phi_1^h), \nabla U(\Phi_1^h)\big)}_{=:R_{t+h}^8} \underbrace{-\sqrt{2}\big(\alpha a_{11}b_1 + \beta a_{21}b_2\big)h\nabla^3 U(x)\big(\nabla U(x), \Delta Z_{t,t+h}\big)}_{=:R_{t+h}^9} \\
& \underbrace{-\sqrt{2}\beta a_{22}b_2 h\nabla^3 U(x)\big(\nabla U(\Phi_1^h), \Delta Z_{t,t+h}\big)}_{=:R_{t+h}^{10}} + \underbrace{\tfrac{\alpha}{2}h\int_0^1 (1-t)^2\nabla^4 U(x + t(\Phi_1^h - x))(\Phi_1^h - x, \Phi_1^h - x, \Phi_1^h - x)\,\mathrm{d}t}_{=:R_{t+h}^{11}} \\
& + \underbrace{\tfrac{\beta}{2}h\int_0^1 (1-t)^2\nabla^4 U(x + t(\Phi_2^h - x))(\Phi_2^h - x, \Phi_2^h - x, \Phi_2^h - x)\,\mathrm{d}t}_{=:R_{t+h}^{12}}.
\end{aligned}
\tag{116}
$$

Before estimating these items, we list some auxiliary lemmas.

**Lemma B.3** (Hölder continuity in time for one-step RKLMC stages). *Let Assumption 2.2 hold and let the timestep satisfy $h \leq 1 \wedge \frac{1}{2L_1} \wedge \frac{1}{2L_1'}$. Then for any $p \geq 1$ and $x \in \mathbb{R}^d$, it holds*

$$\mathbb{E}\left[\big|\Phi_1^h - x\big|^{2p}\right] \leq \left(\mathcal{H}_1^{\Phi,1}(p)d^p + \mathcal{H}_2^{\Phi,1}(p)|x|^{2p}\right)h^p, \tag{117}$$

$$\mathbb{E}\left[\big|\Phi_2^h - x\big|^{2p}\right] \leq \left(\mathcal{H}_1^{\Phi,2}(p)d^p + \mathcal{H}_2^{\Phi,2}(p)|x|^{2p}\right)h^p, \tag{118}$$

*where*

$$
\begin{aligned}
\mathcal{H}_1^{\Phi,1}(p) :=& 2^{3p-2}(a_{11})^{2p}L_1'^p + 3^p(2p-1)!!(b_1)^{2p}, \\
\mathcal{H}_2^{\Phi,1}(p) :=& 2^{3p-2}(a_{11})^{2p}L_1^p, \\
\mathcal{H}_1^{\Phi,2}(p) :=& 3(18)^{p-1}(a_{21})^{2p}L_1'^p + 3(18)^{p-1}(a_{22})^{2p}L_1'^p + 3^{2p}(18)^{p-1}(a_{11}a_{22})^{2p}L_1^p \\
& + 3^{2p}(18)^{p-1}(2p-1)!!(b_1 a_{22})^{2p}L_1^p + 3^{2p-1}(2p-1)!!(b_2)^{2p}, \\
\mathcal{H}_2^{\Phi,2}(p) :=& 3(18)^{p-1}(a_{21})^{2p}L_1^p + 3^{2p}(18)^{p-1}(a_{22})^{2p}L_1^p + 3^{2p}(18)^{p-1}(a_{11}a_{22})^{2p}.
\end{aligned}
\tag{119}
$$

*Proof.* First, we prove the first assertion (117). Recalling (110) and using (23), (54) and the assumption $h \leq 1 \wedge \frac{1}{2L_1} \wedge \frac{1}{2L_1'}$, we take $2p$-norm and expectations on both sides to get

$$
\begin{aligned}
\mathbb{E}\Big[\big|\Phi_1^h - x\big|^{2p}\Big] \leq & 2^{2p-1}(a_{11})^{2p}h^{2p}|\nabla U(x)|^{2p} + 2^{3p-1}(b_1)^{2p}\mathbb{E}\Big[\big|\tfrac{\Delta Z_{t,t+h}}{h}\big|^{2p}\Big] \\
\leq & \Big(2^{4p-2}(a_{11})^{2p}L_1'^{2p}d^p + 2^{4p-2}(a_{11})^{2p}L_1^{2p}|x|^{2p}\Big)h^{2p} + \tfrac{2^{3p-1}}{3^p}(2p-1)!!(b_1)^{2p}d^ph^p \quad (120) \\
\leq & \Big(\big(2^{3p-2}(a_{11})^{2p}L_1'^p + 3^p(2p-1)!!(b_1)^{2p}\big)d^p + 2^{3p-2}(a_{11})^{2p}L_1^p|x|^{2p}\Big)h^p.
\end{aligned}
$$

The first assertion is thus validated. Before coming to the estimate (118), we apply (54) to obtain

$$
\begin{aligned}
\mathbb{E}\Big[\big|\Phi_1^h\big|^{2p}\Big] \leq & 3^{2p-1}|x|^{2p} + 3^{2p-1}(a_{11})^{2p}h^{2p}|\nabla U(x)|^{2p} + 3^{3p-1}(b_1)^{2p}\mathbb{E}\Big[\big|\tfrac{\Delta Z_{t,t+h}}{h}\big|^{2p}\Big] \\
\leq & 3^{2p-1}|x|^{2p} + 6^{2p-1}(a_{11})^{2p}L_1'^{2p}h^{2p}d^p + 6^{2p-1}(a_{11})^{2p}L_1'^{2p}h^{2p}|x|^{2p} + 3^{2p-1}(2p-1)!!(b_1)^{2p}d^ph^p \quad (121) \\
\leq & \mathcal{M}_1^{\Phi,1}(p)d^p + \mathcal{M}_2^{\Phi,1}(p)|x|^{2p},
\end{aligned}
$$

where

$$
\mathcal{M}_1^{\Phi,1}(p) := 3^{2p-1}(a_{11})^{2p} + 3^{2p-1}(2p-1)!!(b_1)^{2p}, \quad \mathcal{M}_2^{\Phi,1}(p) := 3^{2p-1} + 3^{2p-1}(a_{11})^{2p}. \quad (122)
$$

Bearing this in mind and in the same spirit of (118), we acquire

$$
\begin{aligned}
\mathbb{E}\Big[\big|\Phi_2^h - x\big|^{2p}\Big] \leq & 3^{2p-1}(a_{21})^{2p}h^{2p}|\nabla U(x)|^{2p} + 3^{2p-1}(a_{21})^{2p}h^{2p}\mathbb{E}\Big[\big|\nabla U(\Phi_1^h)\big|^{2p}\Big] + 3^{3p-1}(b_2)^{2p}\mathbb{E}\Big[\big|\tfrac{\Delta Z_{t,t+h}}{h}\big|^{2p}\Big] \\
\leq & 6^{2p-1}\big((a_{21})^{2p}L_1'^{2p} + (a_{22})^{2p}L_1'^{2p}\big)d^ph^{2p} + 6^{2p-1}(a_{21})^{2p}L_1^{2p}|x|^{2p}h^{2p} \\
& + 6^{2p-1}(a_{22})^{2p}L_1^{2p}\mathbb{E}\Big[\big|\Phi_1^h\big|^{2p}\Big]h^{2p} + 3^{2p-1}(2p-1)!!(b_2)^{2p}d^ph^p \\
\leq & \Big(3(18)^{p-1}(a_{21})^{2p}L_1'^p + 3(18)^{p-1}(a_{22})^{2p}L_1'^p + 3^{2p}(18)^{p-1}(a_{11}a_{22})^{2p}L_1^p \\
& + 3^{2p}(18)^{p-1}(2p-1)!!(b_1a_{22})^{2p}L_1^p + 3^{2p-1}(2p-1)!!(b_2)^{2p}\Big)d^ph^p \\
& + \Big(3(18)^{p-1}(a_{21})^{2p}L_1^p + 3^{2p}(18)^{p-1}(a_{22})^{2p}L_1^p + 3^{2p}(18)^{p-1}(a_{11}a_{22})^{2p}\Big)|x|^{2p}h^p,
\end{aligned}
$$
$$(123)$$

which implies

$$
\begin{aligned}
\mathcal{H}_1^{\Phi,2}(p) := & 3(18)^{p-1}(a_{21})^{2p}L_1'^p + 3(18)^{p-1}(a_{22})^{2p}L_1'^p + 3^{2p}(18)^{p-1}(a_{11}a_{22})^{2p}L_1^p \\
& + 3^{2p}(18)^{p-1}(2p-1)!!(b_1a_{22})^{2p}L_1^p + 3^{2p-1}(2p-1)!!(b_2)^{2p}, \\
\mathcal{H}_2^{\Phi,2}(p) := & 3(18)^{p-1}(a_{21})^{2p}L_1^p + 3^{2p}(18)^{p-1}(a_{22})^{2p}L_1^p + 3^{2p}(18)^{p-1}(a_{11}a_{22})^{2p},
\end{aligned}
$$
$$(124)$$

as required. $\qquad\square$

The preceding preparations enable us to quantify the one-step weak and strong errors between the TELMC and RKLMC schemes, as stated in the following lemma.

**Lemma B.4.** *Let Assumptions 2.1-2.4 hold and let the timestep $h$ satisfy $h \leq 1 \wedge \frac{1}{2L_1'} \wedge \frac{1}{2L_1}$. Then for any $x \in \mathbb{R}^d$ and any $t \geq 0$, it holds*

$$
\begin{aligned}
\Big|\mathbb{E}\Big[\mathbb{Y}(t,x;t+h) - Y(t,x;t+h)\Big]\Big| \leq & \Big(K_{2,1}^E d^3 + K_{2,2}^E |x|^6\Big)^{\frac{1}{2}} h^{\frac{5}{2}}, \\
\Big(\mathbb{E}\Big[\big|\mathbb{Y}(t,x;t+h) - Y(t,x;t+h)\big|^2\Big]\Big)^{1/2} \leq & \Big(K_{2,1}^M d^3 + K_{2,2}^M |x|^6\Big)^{1/2} h^2,
\end{aligned}
$$
$$(125)$$

*where*

$$
\begin{aligned}
K_{2,1}^{E} :=&\, \beta^2(a_{22})^2\big(\mathcal{H}_1^{\Phi,1}(1) + \mathcal{H}_2^{\Phi,1}(1)\big)L_1^4 + 4\big(\alpha^2(a_{11})^4 + \beta^2(a_{21})^4\big)\big(L_1'^4 + L_1^4\big)L_2^2 + 4(\beta a_{21}a_{22})^2\big(2L_1'^4 + L_1^4 \\
&+ \mathcal{M}_1^{\Phi,1}(2)L_1^4 + \mathcal{M}_2^{\Phi,1}(2)L_1^4\big)L_2^2 + 2\beta^2(a_{22})^4\big(L_1'^4 + \mathcal{M}_1^{\Phi,1}(2)L_1^4 + \mathcal{M}_2^{\Phi,1}(2)L_1^4\big)L_2^2 \\
&+ 2\sqrt{3}(\beta a_{22}b_2)^2\big(L_1'^2 + 3\mathcal{M}_1^{\Phi,1}(1)L_1^2 + 3\mathcal{M}_2^{\Phi,1}(1)L_1^2\big)L_2^2 + \tfrac{\alpha^2}{20}\mathcal{H}_1^{\Phi,1}(3)L_3^2 + \tfrac{\beta^2}{20}\mathcal{H}_1^{\Phi,2}(3)L_3^2,
\end{aligned}
$$

$$
\begin{aligned}
K_{2,2}^{E} :=&\, \beta^2(a_{22})^2\mathcal{H}_2^{\Phi,1}(1)L_1^4 + 4\big(\alpha^2(a_{11})^4 + \beta^2(a_{21})^4\big)L_1^4L_2^2 + 4(\beta a_{21}a_{22})^2\big(1 + \mathcal{M}_2^{\Phi,1}(2)\big)L_1^4L_2^2 \\
&+ 2\beta^2(a_{22})^4\mathcal{M}_2^{\Phi,1}(2)L_2^2 + 6\sqrt{3}(\beta a_{22}b_2)^2\mathcal{M}_2^{\Phi,1}(1)L_1^2L_2^2 + \tfrac{\alpha^2}{20}\mathcal{H}_2^{\Phi,1}(3)L_3^2 + \tfrac{\beta^2}{20}\mathcal{H}_2^{\Phi,2}(3)L_3^2,
\end{aligned}
$$

$$
\begin{aligned}
K_{2,1}^{M} :=&\, 9\Big(2L_2^2 + \beta^2(a_{22})^2\big(\mathcal{H}_1^{\Phi,1}(1) + \mathcal{H}_2^{\Phi,1}(1)\big)L_1^4 + 4\big(\alpha^2(a_{11})^4 + \beta^2(a_{21})^4\big)\big(L_1'^4 + L_1^4\big)L_2^2 \\
&+ 4(\beta a_{21}a_{22})^2\big(2L_1'^4 + L_1^4 + \mathcal{M}_1^{\Phi,1}(2)L_1^4 + \mathcal{M}_2^{\Phi,1}(2)L_1^4\big)L_2^2 + 2\beta^2(a_{22})^4\big(L_1'^4 + \mathcal{M}_1^{\Phi,1}(2)L_1^4 + \mathcal{M}_2^{\Phi,1}(2)L_1^4\big)L_2^2 \\
&+ \tfrac{4}{3}\big(\alpha a_{11}b_1 + \beta a_{21}b_2\big)^2\big(L_1'^2 + L_1^2\big)L_2^2 + 2\sqrt{3}(\beta a_{22}b_2)^2\big(L_1'^2 + 3\mathcal{M}_1^{\Phi,1}(1)L_1^2 + 3\mathcal{M}_2^{\Phi,1}(1)L_1^2\big)L_2^2 \\
&+ \tfrac{\alpha^2}{20}\mathcal{H}_1^{\Phi,1}(3)L_3^2 + \tfrac{\beta^2}{20}\mathcal{H}_1^{\Phi,2}(3)L_3^2\Big),
\end{aligned}
$$

$$
\begin{aligned}
K_{2,2}^{M} :=&\, 9\Big(\beta^2(a_{22})^2\mathcal{H}_2^{\Phi,1}(1)L_1^4 + 4\big(\alpha^2(a_{11})^4 + \beta^2(a_{21})^4\big)L_1^4L_2^2 + 4(\beta a_{21}a_{22})^2\big(1 + \mathcal{M}_2^{\Phi,1}(2)\big)L_1^4L_2^2 \\
&+ 2\beta^2(a_{22})^4\mathcal{M}_2^{\Phi,1}(2)L_2^2 + \tfrac{4}{3}\big(\alpha a_{11}b_1 + \beta a_{21}b_2\big)^2L_1^2L_2^2 + 6\sqrt{3}(\beta a_{22}b_2)^2\mathcal{M}_2^{\Phi,1}(1)L_1^2L_2^2 \\
&+ \tfrac{\alpha^2}{20}\mathcal{H}_2^{\Phi,1}(3)L_3^2 + \tfrac{\beta^2}{20}\mathcal{H}_2^{\Phi,2}(3)L_3^2\Big),
\end{aligned}
\tag{126}
$$

*and constants $\mathcal{H}_j^{\Phi,i}(p)$ and $\mathcal{M}_j^{\Phi,1}(p)$, $i = 1, 2$, $j = 1, 2$, $p \geq 1$, come from* (119) *and* (122)*, respectively.*

*Proof.* Based on the one-step discrepancy between TELMC and RKLMC given in (116), we first apply the triangle inequality to obtain

$$
\Big|\mathbb{E}\Big[\mathbb{Y}(t, x; t + h) - Y(t, x; t + h)\Big]\Big| \leq \sum_{i=4}^{12}\Big|\mathbb{E}\big[R_{t+h}^i\big]\Big|.
\tag{127}
$$

By invoking the fundamental inequality (54), we further derive

$$
\mathbb{E}\Big[\big|\mathbb{Y}(t, x; t + h) - Y(t, x; t + h)\big|^2\Big] \leq 9\sum_{i=4}^{12}\mathbb{E}\Big[\big|R_{t+h}^i\big|^2\Big].
\tag{128}
$$

We next establish bounds on the expectations and second moments of the terms $R_{t+h}^i$, $i = 4, \cdots, 12$, by first estimating their second moments and then deriving the corresponding expectation bounds.

**Estimate of $R_{t+h}^4$:** Before proceeding further, we first employ the Fubini theorem to arrive at

$$
Z_{t,t+h}^l = \int_t^{t+h}\int_s^{t+h} \mathrm{d}r\, \mathrm{d}W_s^l, \quad l \in [d],
\tag{129}
$$

which directly implies

$$
\mathbb{E}\big[Z_{t,t+h}^l\big] = 0, \quad \mathbb{E}\big[|Z_{t,t+h}^l|^2\big] = \tfrac{h^3}{3}.
$$

This together with (44), (46) and (54) implies

$$
\begin{aligned}
\mathbb{E}\Big[\big|R_{t+h}^4\big|^2\Big] \leq&\, \tfrac{9}{h^2}\mathbb{E}\Big[\big|\nabla^3 U(x)\big(\Delta Z_{t,t+h}, \Delta Z_{t,t+h}\big)\big|^2\Big] + \mathbb{E}\Big[\big|\nabla(\Delta U(x))\big|^2\Big]h^4 \\
\leq&\, \tfrac{9L_2^2}{h^2}\mathbb{E}\Big[\big|\Delta Z_{t,t+h}\big|^4\Big] + L_2^2d^2h^4 \\
\leq&\, 2L_2^2d^2h^4 \\
\leq&\, 2L_2^2d^3h^4.
\end{aligned}
\tag{130}
$$

With regard to $|\mathbb{E}[R_{t+h}^4]|$, we will analyze this term component-wisely. Taking expectation on $(R_{t+h}^4)^{(l)}$, $l \in [d]$, shows

$$
\begin{aligned}
\mathbb{E}\Big[(R_{t+h}^4)^{(l)}\Big] =&\, \mathbb{E}\Big[\tfrac{3}{2h}\big(\nabla^3 U(x)\big(\Delta Z_{t,t+h}, \Delta Z_{t,t+h}\big)\big)^{(l)} - \tfrac{h^2}{2}\big(\nabla(\Delta U(x))\big)^{(l)}\Big] \\
=&\, \tfrac{3}{2h}\mathbb{E}\Big[\big(\nabla^3 U(x)\big(\Delta Z_{t,t+h}, \Delta Z_{t,t+h}\big)\big)^{(l)}\Big] - \tfrac{h^2}{2}\big(\nabla(\Delta U(x))\big)^{(l)}.
\end{aligned}
\tag{131}
$$

Noting

$$\mathbb{E}\Big[\Delta Z_{t,t+h}^i \Delta Z_{t,t+h}^j\Big] = \begin{cases} 0 & \text{if } i \neq j, \\ \frac{h^3}{3} & \text{if } i = j. \end{cases} \quad i,j \in [d], \tag{132}$$

and recalling (6), we show that, for $l \in [d]$,

$$\begin{aligned}
\tfrac{3}{2h}\mathbb{E}\Big[\big(\nabla^3 U(x)(\Delta Z_{t,t+h}, \Delta Z_{t,t+h})\big)^{(l)}\Big] &= \tfrac{3}{2h}\mathbb{E}\Big[\sum_{l_1=1}^{d}\sum_{l_2=1}^{d}\tfrac{\partial^3 U(x)}{\partial x_l \partial x_{l_1}\partial x_{l_2}}\Delta Z_{t,t+h}^{l_1}\Delta Z_{t,t+h}^{l_2}\Big] \\
&= \tfrac{3}{2h}\sum_{l_1=1}^{d}\sum_{l_2=1}^{d}\tfrac{\partial^3 U(x)}{\partial x_l \partial x_{l_1}\partial x_{l_2}}\mathbb{E}\Big[\Delta Z_{t,t+h}^{l_1}\Delta Z_{t,t+h}^{l_2}\Big] \\
&= \tfrac{h^2}{2}\sum_{l_1=1}^{d}\tfrac{\partial^3 U(x)}{\partial x_l \partial x_{l_1}\partial x_{l_1}}.
\end{aligned} \tag{133}$$

Moreover, by definitions of the gradient and the Laplacian, we have

$$\tfrac{h^2}{2}\big(\nabla(\Delta U(x))\big)^{(l)} = \tfrac{h^2}{2}\sum_{l_1=1}^{d}\tfrac{\partial^3 U(x)}{\partial x_l \partial x_{l_1}\partial x_{l_1}}. \tag{134}$$

Putting (133) and (134) together, it follows from (131) that

$$\big|\mathbb{E}\big[R_{t+h}^4\big]\big| = 0. \tag{135}$$

**Estimate of $R_{t+h}^5$:** In view of Lemma B.3, we apply (43) and (98) to get

$$\begin{aligned}
\mathbb{E}\Big[\big|R_{t+h}^5\big|^2\Big] &\leq \beta^2(a_{22})^2 L_1^2 h^4 \mathbb{E}\Big[\big|\nabla U(\Phi_1^h) - \nabla U(x)\big|^2\Big] \\
&\leq \beta^2(a_{22})^2 L_1^4 h^4 \mathbb{E}\Big[\big|\Phi_1^h - x\big|^2\Big] \\
&\leq \big(\beta^2(a_{22})^2 \mathcal{H}_1^{\Phi,1}(1) L_1^4 d + \beta^2(a_{22})^2 \mathcal{H}_2^{\Phi,1}(1) L_1^4 |x|^2\big) h^5 \\
&\leq \big(\mathcal{M}_1^{R,5} d^3 + \mathcal{M}_2^{R,5}|x|^6\big) h^5,
\end{aligned} \tag{136}$$

where

$$\mathcal{M}_1^{R,5} := \beta^2(a_{22})^2\big(\mathcal{H}_1^{\Phi,1}(1) + \mathcal{H}_2^{\Phi,1}(1)\big) L_1^4, \quad \mathcal{M}_2^{R,5} := \beta^2(a_{22})^2 \mathcal{H}_2^{\Phi,1}(1) L_1^4. \tag{137}$$

Similar to (102), we arrive at

$$\big|\mathbb{E}\big[R_{t+h}^5\big]\big| \leq \big(\mathcal{M}_1^{R,5} d^3 + \mathcal{M}_2^{R,5}|x|^6\big)^{\frac{1}{2}} h^{\frac{5}{2}}. \tag{138}$$

**Estimate of $R_{t+h}^6$:** Thanks to the linear growth condition (23), (44) and (98), we have

$$\begin{aligned}
\mathbb{E}\Big[\big|R_{t+h}^6\big|^2\Big] &\leq \big(\tfrac{\alpha}{2}(a_{11})^2 + \tfrac{\beta}{2}(a_{21})^2\big)^2 L_2^2 h^6 \big|\nabla U(x)\big|^4 \\
&\leq \big(4\big(\alpha^2(a_{11})^4 + \beta^2(a_{21})^4\big)L_1'^4 L_2^2 d^2 + 4\big(\alpha^2(a_{11})^4 + \beta^2(a_{21})^4\big)L_1^4 L_2^2 |x|^4\big) h^6 \\
&\leq \big(\mathcal{M}_1^{R,6} d^3 + \mathcal{M}_2^{R,6}|x|^4\big) h^6,
\end{aligned} \tag{139}$$

where

$$\begin{aligned}
\mathcal{M}_1^{R,6} &:= 4\big(\alpha^2(a_{11})^4 + \beta^2(a_{21})^4\big)\big(L_1'^4 + L_1^4\big)L_2^2, \\
\mathcal{M}_2^{R,6} &:= 4\big(\alpha^2(a_{11})^4 + \beta^2(a_{21})^4\big)L_1^4 L_2^2.
\end{aligned} \tag{140}$$

In the same spirit as (102), one can derive

$$\big|\mathbb{E}\big[R_{t+h}^6\big]\big| \leq \Big(\mathcal{M}_1^{R,6}d^3 + \mathcal{M}_2^{R,6}|x|^6\Big)^{\frac{1}{2}}h^3. \tag{141}$$

**Estimate of $R_{t+h}^7$:** By an argument analogous to (139), together with (121), we obtain

$$\begin{aligned}
\mathbb{E}\Big[\big|R_{t+h}^7\big|^2\Big] \leq &(\beta a_{21}a_{22})^2 L_2^2 h^6 |\nabla U(x)|^2 \mathbb{E}\Big[\big|\nabla U(\Phi_1^h)\big|^2\Big]\\
\leq &\tfrac{1}{2}(\beta a_{21}a_{22})^2 L_2^2 h^6 |\nabla U(x)|^4 + \tfrac{1}{2}(\beta a_{21}a_{22})^2 L_2^2 h^6 \mathbb{E}\Big[\big|\nabla U(\Phi_1^h)\big|^4\Big]\\
\leq &\Big(4(\beta a_{21}a_{22})^2 L_1'^4 L_2^2 d^2 + 4(\beta a_{21}a_{22})^2 L_1^4 L_2^2 |x|^4\Big)h^6\\
&+ \Big(4(\beta a_{21}a_{22})^2 L_1'^4 L_2^2 d^2 + 4(\beta a_{21}a_{22})^2 L_1^4 L_2^2 \mathbb{E}\big[|\Phi_1^h|^4\big]\Big)h^6\\
\leq &\Big(\big(8(\beta a_{21}a_{22})^2 L_1'^4 L_2^2 + 4(\beta a_{21}a_{22})^2 \mathcal{M}_1^{\Phi,1}(2)L_1^4 L_2^2\big)d^2\\
&+ 4(\beta a_{21}a_{22})^2\big(1 + \mathcal{M}_2^{\Phi,1}(2)\big)L_1^4 L_2^2 |x|^4\Big)h^6\\
\leq &\Big(\mathcal{M}_1^{R,7}d^3 + \mathcal{M}_2^{R,7}|x|^6\Big)h^6,
\end{aligned} \tag{142}$$

where

$$\begin{aligned}
\mathcal{M}_1^{R,7} :=& 4(\beta a_{21}a_{22})^2\big(2L_1'^4 + L_1^4 + \mathcal{M}_1^{\Phi,1}(2)L_1^4 + \mathcal{M}_2^{\Phi,1}(2)L_1^4\big)L_2^2,\\
\mathcal{M}_2^{R,7} :=& 4(\beta a_{21}a_{22})^2\big(1 + \mathcal{M}_2^{\Phi,1}(2)\big)L_1^4 L_2^2.
\end{aligned} \tag{143}$$

Invoking the Hölder inequality, we conclude that

$$\big|\mathbb{E}\big[R_{t+h}^7\big]\big| \leq \Big(\mathcal{M}_1^{R,7}d^3 + \mathcal{M}_2^{R,7}|x|^6\Big)^{\frac{1}{2}}h^3. \tag{144}$$

**Estimate of $R_{t+h}^8$:** Following the same arguments as used in (139) and using (121), we have

$$\begin{aligned}
\mathbb{E}\Big[\big|R_{t+h}^8\big|^2\Big] \leq &\tfrac{\beta^2}{4}(a_{22})^4 L_2^2 h^6 \mathbb{E}\Big[\big|\nabla U(\Phi_1^h)\big|^4\Big]\\
\leq &\Big(2\beta^2 (a_{22})^4 L_1'^4 L_2^2 d^2 + 2\beta^2 (a_{22})^4 L_1^4 L_2^2 \mathbb{E}\Big[\big|\Phi_1^h\big|^4\Big]\Big)h^6\\
\leq &\Big(2\beta^2 (a_{22})^4\big(L_1'^4 + \mathcal{M}_1^{\Phi,1}(2)L_1^4\big)L_2^2 d^2 + 2\beta^2 (a_{22})^4 \mathcal{M}_2^{\Phi,1}(2)L_1^4 L_2^2 |x|^4\Big)h^6\\
\leq &\Big(\mathcal{M}_1^{R,8}d^3 + \mathcal{M}_2^{R,8}|x|^6\Big)h^6,
\end{aligned} \tag{145}$$

where

$$\mathcal{M}_1^{R,8} := 2\beta^2 (a_{22})^4\big(L_1'^4 + \mathcal{M}_1^{\Phi,1}(2)L_1^4 + \mathcal{M}_2^{\Phi,1}(2)L_1^4\big)L_2^2, \quad \mathcal{M}_2^{R,8} := 2\beta^2 (a_{22})^4 \mathcal{M}_2^{\Phi,1}(2)L_2^2. \tag{146}$$

Using an approach analogous to that in (102), one can deduce

$$\big|\mathbb{E}\big[R_{t+h}^8\big]\big| \leq \Big(\mathcal{M}_1^{R,8}d^3 + \mathcal{M}_2^{R,8}|x|^6\Big)^{\frac{1}{2}}h^3. \tag{147}$$

**Estimate of $R_{t+h}^9$:** By using (23), (44) and (98), one infers

$$\begin{aligned}
\mathbb{E}\Big[\big|R_{t+h}^9\big|^2\Big] \leq &2(\alpha a_{11}b_1 + \beta a_{21}b_2)^2 L_2^2 h^2 |\nabla U(x)|^2 \mathbb{E}\Big[\big|\Delta Z_{t,t+h}\big|^2\Big]\\
\leq &\tfrac{4}{3}(\alpha a_{11}b_1 + \beta a_{21}b_2)^2 L_2^2\big(L_1'^2 d^2 + L_1^2 d|x|^2\big)h^5\\
\leq &\Big(\mathcal{M}_1^{R,9}d^3 + \mathcal{M}_2^{R,9}|x|^6\big)\Big)h^5,
\end{aligned} \tag{148}$$

where

$$\mathcal{M}_1^{R,9} := \tfrac{4}{3}(\alpha a_{11}b_1 + \beta a_{21}b_2)^2\big(L_1'^2 + L_1^2\big)L_2^2, \quad \mathcal{M}_2^{R,9} := \tfrac{4}{3}(\alpha a_{11}b_1 + \beta a_{21}b_2)^2 L_1^2 L_2^2. \tag{149}$$

Regarding $|\mathbb{E}[R_{t+h}^9]|$, we use $\mathbb{E}[\Delta Z_{t,t+h}] = 0$ to derive

$$
\begin{aligned}
\big|\mathbb{E}\big[R_{t+h}^9\big]\big| &= \sqrt{2}\big(\alpha a_{11}b_1 + \beta a_{21}b_2\big)h\mathbb{E}\Big[\nabla^3 U(x)\big(\nabla U(x), \Delta Z_{t,t+h}\big)\Big] \\
&= \sqrt{2}\big(\alpha a_{11}b_1 + \beta a_{21}b_2\big)h\nabla^3 U(x)\Big(\nabla U(x), \mathbb{E}\big[\Delta Z_{t,t+h}\big]\Big) \\
&= 0.
\end{aligned}
\tag{150}
$$

**Estimate of $R_{t+h}^{10}$:** Employing the Hölder inequality in conjunction with (23), (44), (98) and (121), one can deduce

$$
\begin{aligned}
\mathbb{E}\Big[\big|R_{t+h}^{10}\big|^2\Big] &\leq 2(\beta a_{22}b_2)^2 L_2^2 h^2 \mathbb{E}\Big[\big|\nabla U(\Phi_1^h)\big|^2 \big|\Delta Z_{t,t+h}\big|^2\Big] \\
&\leq 2(\beta a_{22}b_2)^2 L_2^2 h^2 \Big(\mathbb{E}\Big[\big|\nabla U(\Phi_1^h)\big|^4\Big]\mathbb{E}\Big[\big|\Delta Z_{t,t+h}\big|^4\Big]\Big)^{\frac{1}{2}} \\
&\leq 2(\beta a_{22}b_2)^2 L_2^2 \Big(\big(\tfrac{8}{3}L_1'^4 d^4 + \tfrac{8}{3}L_1^4 d^2 \mathbb{E}\big[\big|\Phi_1^h\big|^4\big]\big)\Big)^{\frac{1}{2}} h^5 \\
&\leq 2(\beta a_{22}b_2)^2 L_2^2 \Big(\tfrac{8}{3}\big(L_1'^4 + \mathcal{M}_1^{\Phi,1}(2)L_1^4\big)d^4 + \tfrac{8}{3}\mathcal{M}_2^{\Phi,1}(2)L_1^4 d^2 |x|^4\Big)^{\frac{1}{2}} h^5 \\
&\leq 2\sqrt{3}(\beta a_{22}b_2)^2 L_2^2 \Big(\big(L_1'^2 + 3\mathcal{M}_1^{\Phi,1}(1)L_1^2\big)d^2 + 3\mathcal{M}_2^{\Phi,1}(1)L_1^2 d|x|^2\Big)h^5 \\
&\leq \Big(\mathcal{M}_1^{R,10}d^3 + \mathcal{M}_2^{R,10}|x|^6\Big)h^5,
\end{aligned}
\tag{151}
$$

where we used $\sqrt{\sum_{i=1}^k u_i} \leq \sum_{i=1}^k \sqrt{u_i}$, $u_i \geq 0$, $k \in \mathbb{N}$, and $(\mathcal{M}_i^{\Phi,1}(2))^{\frac{1}{2}} \leq 3\mathcal{M}_i^{\Phi,1}(1)$, $i = 1, 2$, in the fifth step. Here, we denote

$$
\begin{aligned}
\mathcal{M}_1^{R,10} &:= 2\sqrt{3}(\beta a_{22}b_2)^2\big(L_1'^2 + 3\mathcal{M}_1^{\Phi,1}(1)L_1^2 + 3\mathcal{M}_2^{\Phi,1}(1)L_1^2\big)L_2^2, \\
\mathcal{M}_2^{R,10} &:= 6\sqrt{3}(\beta a_{22}b_2)^2 \mathcal{M}_2^{\Phi,1}(1)L_1^2 L_2^2.
\end{aligned}
\tag{152}
$$

Repeating the argument leading to (102), we arrive at

$$
\big|\mathbb{E}\big[R_{t+h}^{10}\big]\big| \leq \Big(\mathcal{M}_1^{R,10}d^3 + \mathcal{M}_2^{R,10}|x|^6\Big)^{\frac{1}{2}} h^{\frac{5}{2}}.
\tag{153}
$$

**Estimate of $R_{t+h}^{11}$:** By using the Hölder inequality and (45), one can derive from Lemma B.3 that

$$
\begin{aligned}
\mathbb{E}\Big[\big|R_{t+h}^{11}\big|^2\Big] &\leq \tfrac{\alpha^2}{4}h^2 \int_0^1 (1-t)^4 \mathbb{E}\Big[\big|\nabla^4 U(x + t(\Phi_1^h - x))(\Phi_1^h - x, \Phi_1^h - x, \Phi_1^h - x)\big|^2\Big]\,\mathrm{d}t \\
&\leq \tfrac{\alpha^2}{4}L_3^2 h^2 \int_0^1 (1-t)^4\,\mathrm{d}t\,\mathbb{E}\Big[\big|\Phi_1^h - x\big|^6\Big] \\
&\leq \Big(\tfrac{\alpha^2}{20}\mathcal{H}_1^{\Phi,1}(3)L_3^2 d^3 + \tfrac{\alpha^2}{20}\mathcal{H}_2^{\Phi,1}(3)L_3^2 |x|^6\Big)h^5.
\end{aligned}
\tag{154}
$$

Similar to (102), we infer

$$
\big|\mathbb{E}\big[R_{t+h}^{11}\big]\big| \leq \Big(\tfrac{\alpha^2}{20}\mathcal{H}_1^{\Phi,1}(3)L_3^2 d^3 + \tfrac{\alpha^2}{20}\mathcal{H}_2^{\Phi,1}(3)L_3^2 |x|^6\Big)^{\frac{1}{2}} h^{\frac{5}{2}}.
\tag{155}
$$

**Estimate of $R_{t+h}^{12}$:** Similar to (154), one can get

$$
\mathbb{E}\Big[\big|R_{t+h}^{12}\big|^2\Big] \leq \Big(\tfrac{\beta^2}{20}\mathcal{H}_1^{\Phi,2}(3)L_3^2 d^3 + \tfrac{\beta^2}{20}\mathcal{H}_2^{\Phi,2}(3)L_3^2 |x|^6\Big)h^5.
\tag{156}
$$

By an argument identical to that used in (102), it holds

$$
\big|\mathbb{E}\big[R_{t+h}^{12}\big]\big| \leq \Big(\tfrac{\beta^2}{20}\mathcal{H}_1^{\Phi,2}(3)L_3^2 d^3 + \tfrac{\beta^2}{20}\mathcal{H}_2^{\Phi,2}(3)L_3^2 |x|^6\Big)^{\frac{1}{2}} h^{\frac{5}{2}}.
\tag{157}
$$

By collecting the preceding bounds, the desired assertion follows immediately from (127) and (128). $\qquad\square$

In conclusion, combining Lemmas B.2 and B.4 and invoking the triangle and fundamental inequalities (54) give the one-step weak and strong error estimates between the Langevin SDE and the RKLMC scheme, summarized as follows.

**Lemma B.5.** *Let Assumptions 2.1-2.4 hold and let the timestep $h$ satisfy $h \leq 1 \wedge \frac{1}{2L_1'} \wedge \frac{1}{2L_1}$. Then for any $x \in \mathbb{R}^d$ and any $t \geq 0$, it holds*

$$\left|\mathbb{E}\big[X(t,x;t+h) - Y(t,x;t+h)\big]\right| \leq \big(K_1^E d^3 + K_2^E |x|^6\big)^{1/2} h^{\frac{5}{2}},$$

$$\left(\mathbb{E}\big[\big|X(t,x;t+h) - Y(t,x;t+h)\big|^2\big]\right)^{1/2} \leq \big(K_1^M d^3 + K_2^M |x|^6\big)^{1/2} h^2, \tag{158}$$

*where*

$$K_i^E = K_{1,i}^E + K_{2,i}^E, \quad K_i^M = 2K_{1,i}^M + 2K_{2,i}^M \quad i = 1, 2, \tag{159}$$

*and constants $K_{1,i}^E$, $K_{1,i}^M$ and $K_{2,i}^E$, $K_{2,i}^M$, $i = 1, 2$, come from (92) and (126), respectively.*

*Proof of Proposition 3.3.* This result is established by applying Theorem 3.3 in (Yang & Wang, 2025). Therefore, it suffices to verify the conditions required by the theorem (i.e., Conditions A1–A4 and Eq. (46) of Theorem 3.3 in (Yang & Wang, 2025)).

Under Assumption 2.1, Condition (A1) is satisfied with

$$\hat{\mu}^* = \mu' d, \quad \mu^* = \mu.$$

From Proposition 3.2, we deduce that Condition (A2) holds with

$$C_1^* = e^{-\frac{\mu}{4}t}, \quad \hat{C}_1^* = \mathcal{M}_2(p)d^p, \quad h_0 = 1 \wedge \frac{1}{2L_1'} \wedge \frac{1}{2L_1} \wedge \frac{4}{\mu} \wedge \frac{\mu}{32L_1^2} \wedge \frac{\mu}{4\kappa_1 L_1^2} \wedge \frac{\mu^2}{8\kappa_1 L_1^3}.$$

Assumption 2.2 ensures that Conditions (A3) and (A4) hold with

$$L^* = L_1, \quad L_f^* = \tfrac{1}{3}L_1, \quad r_0 = 0.$$

By Lemma B.5, Eq. (46) holds with

$$\hat{K}_1^* = K_1^E d^3, \qquad K_1^* = K_2^E, \quad \hat{K}_2^* = K_1^M d^3, \qquad K_2^* = K_2^M, \quad r = 3. \tag{160}$$

Consequently, we obtain

$$\mathbb{E}\big[\big|X_{t_n} - Y_n\big|^2\big] \leq \underbrace{e^{(1+12L_1)T}}_{=:C(T)} \Big(\big(\underbrace{K_1^E + 5K_1^M + \mathcal{M}_2(3)\big(K_2^E + 5K_2^M\big)}_{=:K_1}\big)d^3 + \big(\underbrace{K_2^E + 5K_2^M}_{=:K_2}\big)\mathbb{E}[|X_0|^6]\Big)h^3. \tag{161}$$

The proof is thus finished. $\qquad\square$

# C. Proofs of Theorem 2.6 and Proposition 2.7

*Proof of Theorem 2.6.* We first fix a terminal time $T = n_1 h$ with $n_1 \in \mathbb{N}$. For any $n \geq n_1$, applying the triangle inequality for the $\mathcal{W}_2$-distance yields

$$\mathcal{W}_2\big(\nu q_n, \pi\big) \leq \mathcal{W}_2\big(\nu q_{n-n_1} q_{n_1}, \nu q_{n-n_1} p_{n_1 h}\big) + \mathcal{W}_2\big(\nu q_{n-n_1} p_{n_1 h}, \pi\big). \tag{162}$$

We estimate the two terms on the right-hand side separately. Observe that

$$\mathcal{W}_2\big(\nu q_{n-n_1} q_{n_1}, \nu q_{n-n_1} p_{n_1 h}\big) = \mathcal{W}_2\Big(\mathcal{L}\big(Y(t_{n-n_1}, Y_{n-n_1}; t_n)\big), \mathcal{L}\big(X(t_{n-n_1}, Y_{n-n_1}; t_n)\big)\Big), \tag{163}$$

where $Y(s,x;t)$ and $X(s,x;t)$ denote, respectively, the solutions of the RKLMC scheme (10) and the Langevin SDE (2) at time $t$, initialized from $x$ at time $s$. Invoking Propositions 3.2 and 3.3 yields

$$\mathcal{W}_2^2\Big(\mathcal{L}\big(Y(t_{n-n_1}, Y_{n-n_1}; t_n)\big), \mathcal{L}\big(X(t_{n-n_1}, Y_{n-n_1}; t_n)\big)\Big)$$
$$\leq \mathbb{E}\Big[\big|X(t_{n-n_1}, Y_{n-n_1}; t_n) - Y(t_{n-n_1}, Y_{n-n_1}; t_n)\big|^2\Big]$$
$$\leq \exp\big(1 + 12L_1 T\big)\Big(K_1 d^3 + K_2 \mathbb{E}\Big[\big|Y_{n-n_1}\big|^6\Big]\Big)h^3$$
$$\leq \exp\big(1 + 12L_1 T\big)\Big(\big(K_1 d^3 + K_2 \mathcal{M}_2(3)\big)d^3 + K_2 \mathbb{E}\big[|X_0|^6\big]\Big)h^3, \tag{164}$$

which in turn implies

$$\mathcal{W}_2\big(\nu q_{n-n_1} q_{n_1}, \nu q_{n-n_1} p_{n_1 h}\big) \leq \exp\big(\tfrac{1+12L_1 T}{2}\big)\Big(\big(K_1 d^3 + K_2 \mathcal{M}_2(3)\big)d^3 + K_2 \mathbb{E}\big[|X_0|^6\big]\Big)^{\frac{1}{2}} h^{\frac{3}{2}}. \tag{165}$$

With regard to the second term $\mathcal{W}_2(\nu q_{n-n_1} p_{n_1 h}, \pi)$, using the exponential ergodicity estimate (3.4), we have

$$\mathcal{W}_2\big(\nu q_{n-n_1} p_{n_1 h}, \pi\big) \leq \mathcal{K} e^{-\eta n_1 h} \mathcal{W}_2\big(\nu q_{n-n_1}, \pi\big). \tag{166}$$

For a given stepsize $h > 0$, we choose

$$n_1 = \big\lceil \tfrac{\log \mathcal{K}+1}{\eta h} \big\rceil, \tag{167}$$

for which $n_1$ is an integer. Since $h \leq \frac{1}{2L_1}$, this implies

$$T := n_1 h \leq \big(\tfrac{\log \mathcal{K}+1}{\eta h} + 1\big)h \leq \tfrac{\log \mathcal{K}+1}{\eta} + \tfrac{1}{2L_1} =: \Theta. \tag{168}$$

Moreover,

$$0 < \mathcal{K} e^{-\eta n_1 h} \leq e^{-1} < 1. \tag{169}$$

Combining the above bounds and applying Lemma D.1 of (Yang & Wang, 2025), we arrive at

$$\begin{aligned}
\mathcal{W}_2\big(\nu q_n, \pi\big) &\leq \exp\big(\tfrac{1+12L_1 T}{2}\big)\Big(\big(K_1 + K_2 \mathcal{M}_2(3)\big)d^3 + K_2 \mathbb{E}\big[|X_0|^6\big]\Big)^{\frac{1}{2}} h^{\frac{3}{2}} + \tfrac{1}{e}\mathcal{W}_2\big(\nu q_{n-n_1}, \pi\big) \\
&\leq 2\exp\big(\tfrac{1+12L_1 T}{2}\big)\Big(\big(K_1 + K_2 \mathcal{M}_2(3)\big)d^3 + K_2 \mathbb{E}\big[|X_0|^6\big]\Big)^{\frac{1}{2}} h^{\frac{3}{2}} + e^{1-\frac{n}{n_1}} \sup_{k \in [n_1-1]_0} \mathcal{W}_2\big(\nu q_k, \pi\big).
\end{aligned} \tag{170}$$

Recalling the definition of the $\mathcal{W}2$-distance, together with (54), Lemma 3.1, and Proposition 3.2, we obtain

$$\sup_{k \in [n_1-1]_0} \mathcal{W}_2\big(\nu q_k, \pi\big) \leq \sup_{k \geq 0} \Big(2\mathbb{E}\big[|Y_k|^2\big] + 2\mathbb{E}\big[|X_{t_k}|^2\big]\Big)^{\frac{1}{2}} \leq \Big(2\big(\mathcal{M}_1(1) + \mathcal{M}_2(1)\big)d + 4\mathbb{E}\big[|X_0|^2\big]\Big)^{\frac{1}{2}}. \tag{171}$$

Finally, by (167), we have

$$\frac{n}{n_1} \geq \frac{n}{\frac{\log \mathcal{K}+1}{\eta h}+1} \geq \frac{\eta n h}{\log \mathcal{K}+1+\eta/(2L_1)}. \tag{172}$$

Denoting

$$\lambda := \frac{\eta}{\log \mathcal{K}+1+\eta/(2L_1)}, \tag{173}$$

it follows that $e^{-n/n_1} \leq e^{-\lambda n h}$ and thus

$$\begin{aligned}
\mathcal{W}_2\big(\nu q_n, \pi\big) &\leq 2\exp\big(\tfrac{1+12L_1 \Theta}{2}\big)\Big(\big(K_1 + K_2 \mathcal{M}_2(3)\big)d^3 + K_2 \mathbb{E}\big[|X_0|^6\big]\Big)^{\frac{1}{2}} h^{\frac{3}{2}} \\
&\quad + e\Big(2\big(\mathcal{M}_1(1) + \mathcal{M}_2(1)\big)d + 4\mathbb{E}\big[|X_0|^2\big]\Big)^{\frac{1}{2}} e^{-\lambda n h} \\
&\leq \underbrace{2\exp\big(\tfrac{1+12L_1 \Theta}{2}\big)\big(K_1 + K_2 \mathcal{M}_2(3) + K_2 \sigma(3)\big)^{\frac{1}{2}}}_{=:C_1} d^{\frac{3}{2}} h^{\frac{3}{2}} \\
&\quad + \underbrace{e\big(2\mathcal{M}_1(1) + 2\mathcal{M}_2(1) + 4\sigma(1)\big)^{\frac{1}{2}}}_{=:C_2} d^{\frac{1}{2}} e^{-\lambda n h},
\end{aligned} \tag{174}$$

where in the last step we used $\mathbb{E}[|X_0|^{2q}] \leq \sigma(q)d^q$, for any $q \leq 3$. This completes the proof. $\square$

*Proof of Proposition 2.7.* Given an error tolerance $\epsilon > 0$, by virtue of Theorem 2.6, one can choose $k$ to be large enough and $h$ to be small enough such that

$$C_1 d^{\frac{3}{2}} h^{\frac{3}{2}} \leq \tfrac{\epsilon}{2}, \quad C_2 d^{\frac{1}{2}} e^{-\lambda n h} \leq \tfrac{\epsilon}{2}, \tag{175}$$

which guarantees

$$W_2\big(\nu q_n, \pi\big) \leq \epsilon. \tag{176}$$

We can solve the first term of inequality (175) to ensure

$$\frac{1}{h} \geq \frac{(2C_1)^{\frac{2}{3}} d}{\epsilon^{\frac{2}{3}}}. \tag{177}$$

Then the second part of inequality (175) holds on the condition

$$n \geq \frac{1}{\lambda h} \log \left( \frac{2C_2 \sqrt{d}}{\epsilon} \right). \tag{178}$$

Inserting (177) into (178) yields

$$n \geq \frac{1}{\lambda} \cdot \frac{(2C_1)^{\frac{2}{3}} d}{\epsilon^{\frac{2}{3}}} \cdot \log \left( \frac{2C_2 \sqrt{d}}{\epsilon} \right) = \tilde{O}\big(d\epsilon^{-\frac{2}{3}}\big), \tag{179}$$

as required. $\square$

