# OpenReview forum: "Accelerating Langevin Monte Carlo via Efficient Stochastic Runge-Kutta Methods beyond Log-Concavity"
_ICML.cc/2026/Conference — ICML 2026 regular_

### Official Review · Reviewer_8nPo · 2026-02-26

**Soundness:** 4
**Presentation:** 4
**Significance:** 3
**Originality:** 3
**Overall Recommendation:** 5
**Confidence:** 4

**Summary:**

In this paper the authors consider a novel Runge-Kutta discretization of the Langevin dynamics for sampling from a distribution $\pi\propto e^{-U(x)}$. The scheme builds on an Ito-Taylor approximation of the Langevin SDE. In order to avoid the computations of derivatives of order higher than 1 of $U$, within the scheme a multi-stage update is performed where the coefficients are chosen in such a way, that they match the conventional approximation of the higher-order Ito-Taylor approx. Thus, higher order convergence bounds are achieved without the necessity of computing higher order derivatives.

The authors also provide a small set of synthetic experiments comparing to related methods.

**Compliance With Llm Reviewing Policy:**

Affirmed.

**Final Justification:**

Technically strong paper which proves rigorously the convergence of higher-order Runge-Kutta type schemes for Langevin sampling. Convergence relies on "matching" coefficients of the Ito-Taylor expansion. The higher-order scheme yields higher-order convergence without the same order of derivative computations.

Overall good paper based on technical soundness, originality, and novelty.

**Key Questions For Authors:**

1. In Table 1 the authors write that the M-T of TELMC is worse than that of RKLMC. This seems odd. The proof presented in the paper estimates the discretization error via bounding the errors between TELMC and the continuous-time dynamics and TELMC and RKLMC and afterwards applying the triangle inequality. Thus, I would argue that the complexity of TELMC cannot be worse than that of RKLMC.

2. The numerical experiments are not properly explained. It is not clear how exactly the errors are computed. It sounds as if the authors would focus on the discretization errors? Please clarify this significantly. In particular, since the target is a GMM, exact samples are accessible, so that also for instance one could quantify convergence by evaluating the Wasserstein distance of marginals.

3. Smaller errors/remarks:
* Maybe mention the invoking of Lemma B.1 within equation (97)
* line 810 is duplicate
* I think in line 929 the Brownian motion term ($\Delta W$) is missing, same in equation 115
* in line 1004 the last term lacks a $d^p$ I believe
* In the statement of Lemma B.4 in line 1041 the $X$ should be a $Y$.
* in lie 1088 the second term should be positive
* in line 1132 I think the dimesnion is missing
* in line 1358 at the end $e{-\lambda n h}$ is missing

**Limitations:**

The authors have not addressed any limitations. I think it would be worth discussing limitations and potential remedies regarding maybe the strong assumptions on the potential.

**Strengths And Weaknesses:**

Soundness:
The paper is technically sound. I went through all the proofs and did not find any mistakes. For some smaller errors, see below.

Presentation:
The paper is well-presented and clearly written. I had no issues following the content.

Significance:
The paper is moderately significant. One hand-side, sampling in high dimensions is notoriously time consuming so that acceleration techniques are without a doubt relevant. However, the significance might be limited due to rather strong assumptions on the potential, which are necessary by nature of the higher-order Ito-Taylor approach. In addition, the numerical validation is not very extensive.

Originality:
The paper is fairly original. The main ideas strongly build on ideas from Runge-Kutta discretizations in the context of ODEs and the estimates are mostly quite standard, but I believe overall collecting the results and putting everything together, including the deliberate choice of the parameters to achieve two gradient evaluations, is sufficiently original.

---

> ### Author Rebuttal · Authors · 2026-03-31
>
> We sincerely thank the reviewer for the thorough evaluation and the positive assessment of the soundness, clarity and originality for this paper. We will carefully address each of the raised questions and comments in a detailed, point-by-point manner below.
>
> **About *Weakness***
> > However, the significance ....
>
> **Response:** Thanks a lot for your comments. As you mentioned, to get higher convergence order one needs strong smoothness assumptions by nature of the higher-order Ito-Taylor approach. This could be possibly remedied by introducing the idea of randomized schemes, which is on the list of our future projects. In addition, we will present more numerical experiments, as also suggested by other referees.
>
> **About *Key Questions For Authors***
>
> **Response to Q1:** Thanks for your question.
>
> Please kindly note that the final error bound is influenced by two components: (i) the uniform moment bound of the numerical solution (Proposition 3.2); (ii) the local discretization error (Lemma B.5). While the triangle inequality suggests that TELMC should not be worse than RKLMC at the level of local error, the uniform moment bound of TELMC has  a worse dependence on the dimension $d$ than that of RKLMC, which leads to different error bounds.
>
> More specifically, for TELMC, the presence of higher-order terms involving $\nabla^2 U$ and $\nabla (\Delta U)$ leads to a stronger dependence on the dimension $d$ in the uniform moment bound (see Proposition 3 in Sabanis and Zhang (2019)), i.e. ,
> $$E [ | \mathbb{Y}_n |^{2p}]\leq e^{-c t_n}E [ | X_0 |^{2p} ]+C d^{2p},$$
> where $\mathbb{Y}_n $ denote the TELMC algorithm and $c, C \geq 0 $ are two constants.
> Combining this with Lemma B.2 in our paper and applying Theorem 3.3 of (Yang and Wang, ICML 2025), the resulting finite-time error bound scales as $O(d^2 h^{3/2})$. This worsened dimension dependence ultimately leads to a larger mixing time for TELMC.
>
> We will incorporate this discussion into the revised manuscript to improve its clarity and completeness.
>
>  **Response to Q2:**  Thanks for your helpful comments and sorry for not clarifying this clearly.  The mean-square error refers to the square of $L^2$-error:$(\mathbb{E}|Y_T^{h}-X_T^{ref}|^2)^{1/2}$, where $Y_T^{h}$ is the terminal value produced by a numerical scheme with stepsize $h$ and $X_T^{ref}$ is the reference ("exact") solution to the SDE. The reference solution is taken to be the finest-grid LMC trajectory with stepsize $h_{ref}=2^{-15}$, while the LMC, SRK-2G and SRK-LD schemes are tested on stepsizes $h=2^{-2},\cdots, 2^{-6}$. All methods are driven by the same pre-generated Brownian increments through a common-random-number coupling, so that the terminal-time root mean-square errors (RMSE) can be compared fairly.
> We used $5000$ Monte Carlo samples to approximate the expectation. All detailed settings will be clarified in the revision.
>
> As you mentioned, the considered errors are the discretization errors, which are important as we propose new RK schemes and their mean-square convergence rates need to be numerically confirmed. In addition, our methodology essentially relies on such a mean-square error. Following you suggestion, we quantify convergence by evaluating the Wasserstein distance of marginals for GMM and will add related results in the revised manuscript.
>
>  **Response to Q3:** We sincerely thank the reviewer for the careful reading and detailed verification of the manuscript. The issues you identified are indeed important and your suggestions are very helpful.
> We will revise the paper accordingly by addressing each of the listed points one by one, including:
> - clarifying the use of Lemma B.1,
> - removing the duplicated line 810,
> - adding the missing Brownian motion terms $(\Delta W)$,
> - adding the missing term $d^p$,
> - correcting the notation in Lemma B.4,
> - changing the incorrect negative sign to a positive one,
> - adding the missing term $d$,
> - restoring the missing term $e^{-\lambda n h}$.
>
> These corrections will substantially improve the accuracy and clarity of the manuscript.
>
> **About *Limitations***
>
> **Response:** We fully agree that it would be worth discussing limitations and potential remedies regarding maybe the strong assumptions on the potential.
>
> In this revision, we will add some discussion on the limitations and potential remedies.
> As also noted in your previous comment: “the significance might be limited due to rather strong assumptions on the potential, which are necessary by nature of the higher-order Ito-Taylor approach.”
> For higher-order LMC algorithms, requiring the third derivative of $U$ to be Lipschitz continuous—although commonly adopted in works such as Li et al. (2019) and Sabanis and Zhang (2019)—remains a strong assumption.
>
> To address this limitation, randomized techniques may provide a promising avenue to relax the smoothness requirements and potentially improve the dimension dependence, which we view as an interesting direction for our future work.

---

> > ### Author Rebuttal · Reviewer_8nPo · 2026-04-01
> >
> > Authors resolved all my concerns.

---

### Official Review · Reviewer_7q7p · 2026-03-02

**Soundness:** 4
**Presentation:** 3
**Significance:** 3
**Originality:** 3
**Overall Recommendation:** 5
**Confidence:** 4

**Summary:**

The authors introduce a novel Runge-Kutta based Langevin algorithm to deal with the problem of sampling from non-logconcave potentials.

The new higher-order Langevin algorithm requires two gradient evaluations and achieves Wasserstein-2 accuracy of $\mathcal{O}\left( h^\frac{3}{2} d^{\frac{3}{2}} under dissipativity and gradient and Hessian Lipschitz-smooth assumptions.

In contrast to the existing Runge-Kutta type LMC (Li et al.,2019) involved with three gradient evaluations, the newly proposed algorithm is computationally cheaper and the authors manage to obtain the same convergence guarantees without assuming log-concavity.

**Compliance With Llm Reviewing Policy:**

Affirmed.

**Final Justification:**

This is an interesting, original and technically sound article. The authors have answered all my questions, therefore a give a final score of 5.

**Key Questions For Authors:**

1. There is another article that involves high-order algorithms that also works under assumptions beyond convexity
Neufeld, Ariel, and Ying Zhang. "Non-asymptotic estimates for accelerated high order Langevin Monte Carlo algorithms." arXiv preprint arXiv:2405.05679 (2024).
I believe that I comparison needs to be made with this.

2. In the article a Lipschitz condition is assumed on the Hessian. Such assumptions are standard in high-order schemes. Could you please clarify the points where this is needed in the proof?

3.  Is it possible that one is able to relax this assumption to a Hoeder assumption on the Hessian, or even for the gradient?

**Limitations:**

Yes

**Strengths And Weaknesses:**

Soundness: The article is based on strong foundations, the arguments are presented in a clear way. By first proving uniform in time moment bounds for the new algorithm the authors are able to compute the strong and weak error one step erros which lead to finite time Wasserstein error between the algorithm and the SDE started at the same initial condition (a previous step of the algorithm). In addition, by combining the Harnack inequality implied by gradient smoothness and the LSI they are able provide contraction towards the invariant measure.
Putting all together in eq (28) they are able to provide convergence guarantees from the algorithm to the invariant measure.

Presentation: The presenatation is clear, the authors present the algorithm in a clear way and also provide a clear proof roadmap for the reader.

Significance: The paper is  significant in the sense that it explores high-order Langevin algorithms in depth and provides state of the art guarantees under a non-logconcave framework using a computationally cheaper algorithm.

Originality: The originality of the paper stems from the fact that are new algorithm is presented providing an alternative scheme that does not involve the Hessian and relies only on two gradient evaluations.

---

> ### Author Rebuttal · Authors · 2026-03-30
>
> We are grateful to the reviewer for the careful reading and constructive comments. We will address all the raised questions one by one with detailed clarifications below. We appreciate these valuable suggestions, which will substantially improving the presentation of the paper.
>
> **About *Key Questions For Authors***
> >1. There is another article ... comparison needs to be made with this.
>
> **Response:** We sincerely appreciate the reviewer for drawing our attention to this relevant work, which is also devoted to accelerated Langevin Monte Carlo methods in non-convex settings.
> A detailed comparison with Neufeld and Zhang (2024) will be added in the revision. Next, we summarize main points of comparison below.
>
> - *Different algorithms*. The aim of Neufeld and Zhang (2024) is mainly to sample from target distributions with possibly super-linearly growing potentials. To handle this, the taming strategy is applied to an order $1.5$ Taylor scheme via the Ito-Taylor expansion, involving the Hessian. Instead, this paper focus on the  gradient Lipschitz setting and aims to introduce Hessian-free LMC sampling algorithms based efficient stochastic Runge-Kutta methods.
>
> - *Different methodology and different convergence rates*. The main result of Neufeld and Zhang (2024) is an error bound in $W_1$ distance. As a consequence of it, the authors also provided an error bound in $W_2$ distance, but with reduced convergence rate. Instead, we used different methodology to analyze the error bound in $W_2$ distance directly and obtain an convergence rate of order $1.5$.
>
> - *Different dimension dependence*. The dimension dependence of Neufeld and Zhang (2024) is order $O(e^{O(d)})$ in error bounds. Instead, this paper gives a dimension dependence of order $O(d^{\frac32})$ in error bounds.
>
> In the following table, we present some details on the aforementioned difference.
> | Reference            |Assumptions                                                                 | Distance | Mixing time       |
> |---------------------|---------------------------------------------------------------------------|----------|----------------------------------|
> | Neufeld and Zhang   |  $\nabla^2(\nabla U)^{(i)}$ is $q$-Hölder continuous, $q\in(0,1]$  | W₁       | $\tilde{O}(e^{O(d)} \epsilon^{-{\frac{2}{2+q}}})$     |
> |  Neufeld and Zhang  |    $\nabla^2(\nabla U)^{(i)}$ is $q$-Hölder continuous, $q\in(0,1]$                                                                 |   W₂       | $\tilde{O}(e^{O(d)} \epsilon^{-{\frac{4}{2+q}}})$   |
> | This work           | $\nabla^3 U$ is Lipschitz continuous and $\pi$ satisfies Log-Sobolev inequality            | W₂       | $\tilde{O}(d^{\frac{2}{3}} \epsilon^{-{\frac{2}{3}}})$   |
>
> > 2. In the article ... needed in the proof?
>
> **Response:** Thanks! The Hessian Lipschitz condition is directly used in the estimates of $R_{t+h}^{1,2}$ and $R_{t+h}^{3}$, i.e., in the second step of Eq.(99) and the third step of Eq.(106). Moreover, by Lemma A.1, we obtain a useful consequence of the Hessian Lipschitz condition in Eq.(44), which is further used in the estimates of $R_{t+h}^{i}$ for $i=4,6,7,\cdots,10$.
>
> In the revision, we will clarify it by adding comments.
>
> > 3. Is it possible ... even for the gradient?
>
> **Response:** Thank the reviewer for this very insightful question. Following our proof of the error bounds, we find that, the Hessian Lipschitz condition can be relaxed to a Hölder assumption, but at a cost of severe order reduction. Instead, relaxing the gradient Lipschitz condition to a Hölder assumption would lead to essential difficulties in the error analysis. More precisely, the essential use of the Grownwall inequality in deriving strong convergence theorem (i.e., one-step error to global error) relies on the gradient Lipschitz condition.

---

> > ### Author Rebuttal · Reviewer_7q7p · 2026-04-01
> >
> > The authors have answered all my questions and addressed my concerns.

---

### Official Review · Reviewer_kJTY · 2026-03-12

**Soundness:** 4
**Presentation:** 4
**Significance:** 3
**Originality:** 3
**Overall Recommendation:** 5
**Confidence:** 4

**Summary:**

This paper presents a new approach to Langevin Monte Carlo by discretizing the Langevin stochastic differential equation using a Runge--Kutta scheme. Crucially, the approach does not rely on higher-order derivative information but still achieves a strong order of 1.5 for overdamped Langevin dynamics. Moreover, compared to previous approaches that relied on three gradient evaluations per iteration, the authors propose an approach that only requires two gradient evaluations. The authors also present theory for non-asymptotic error bounds even in non-log-concave settings. The experiments in the paper suggest that the new algorithm is effective and has the correct scaling as expected from the Runge--Kutta discretization.

**Compliance With Llm Reviewing Policy:**

Affirmed.

**Final Justification:**

This paper is technically strong and potentially useful for practitioners. The authors have addressed my concerns in the rebuttal. I have decided to maintain my score of "accept" because I view that there are still some minor gaps to fill in future work despite the overall strength of the paper.

**Key Questions For Authors:**

I have left a few comments above regarding minor typographical/presentation edits. Regarding a few important questions, I would like the authors to additionally address the following points:

1. When reading (17)-(20), an immediate question is: “is there any setting that is ‘best’” in some sense? I think it is worth discussing this further and adding some comments.
2. There are a few places where it is mentioned that a certain constant is “dimension independent” (e.g., Assumption 2.5, Theorem 2.6, etc.) What is exactly meant here? For instance, in Assumption 2.5, $\pi$ is fixed at the beginning of the assumption statement, so what is the sequence of distributions as a function of dimension that you are considering? This causes quite a bit of confusion as it is not rigorously stated as such.
3. Why does RKLMC-2G look like it might even be better than order 1.5 at the top right of Figure 1? Can we do more analysis here to see if this is just an artefact of the experimental settings?

**Limitations:**

Yes

**Strengths And Weaknesses:**

Overall, the theory in the paper seems to be correct and the material is presented well. While I did not check proofs in detail, the overall proof technique, which was also outlined in the main text, seems to be sensible and correct. In terms of presentation, I think that the content was well laid out, with a few comments regarding spelling/grammar that I take note of in more detail at the end of this response. Another comment that I have in terms of presentation is that the main text is a bit short (i.e., 7 pages out of the 8 that were allowed). This is not a hard constraint, but I think that the authors should take advantage of the extra space that is given to them (e.g., include more figures, discussion, etc.)

This paper has original content and is important from two standpoints. First, it presents new uniform-in-time convergence rates in non-log-concave settings for the chosen discretization scheme, which are novel to my knowledge. Second, the Runge--Kutta scheme with two gradient evaluations is new and therefore there is also a practical contribution outside of the theory. The experiments also suggest that the proposed method is useful and has desirable convergence properties.

Here are a few more comments that I think the authors should address:
- The second sentence of the first paragraph in Section 1 contains the word “diverse” twice. It is a bit awkward to read, so please rephrase it.
- In general, Euler--Maruyama, Runge--Kutta etc should be with an “en” dash instead of a hyphen (i.e., -- instead of -)
- In the abstract, “high dimensional” should be replaced with “high-dimensional” since it is being used as an adjective
- The last sentence on page 2 can be rephrased. The “but under the log-concavity condition” is stated in a way that makes it seem that you are the ones assuming log-concavity, rather than Li et al. (2019). This might confuse some readers who quickly read this part.
- In equations (10)-(14), are $\alpha$ and $\beta$ already defined at this point?
- The authors remark that “this reduction would lead to a significant saving in computational cost, especially in high-dimensional settings,” referring to going from 3 gradient evaluations down to two. I don’t see how the savings are larger in high dimensions, since there is a constant saving of a factor of 2/3, which does not depend on dimension. I.e., the percentage compute cost reduction is the same for all choices of $d$.
- Define $\tilde O$ notation explicitly
- Please take a detailed look at the references. There are a few capitalization checks needed. For instance, book titles should generally be capitalized (“Solving Ordinary Differential Equations”), and names (“Poincare”), as well as the “en” dash versus hyphen distinction (Fokker--Planck). Journals/conferences should also be capitalized (“Advances in Neural Information Processing Systems"), etc.

---

> ### Author Rebuttal · Authors · 2026-03-31
>
> We thank the reviewer for this review. All comments/suggestions will be carefully taken into account when revising the manuscript. Detailed responses to each comment are provided below.
>
> **About *Weaknesses***
> > Another comment ... .
>
> **Response:** Many thanks! As you and other referees suggested, we will spend the left space for two additional numerical examples and some discussions on, e.g., assumptions, results and methods.
>
> >Here are  ... .
>
> **Response:**  Thank you for your careful review and insightful suggestions. We will revise the following issues one by one.
> -  We replace the first “diverse” with "many".
> - Done
> - Done
> - We rephrase it to avoid any misunderstanding.
> - Here, $\alpha$ and $\beta$ should be arbitrary real numbers, i.e., **$\alpha, \beta \in \mathbb{R}.$**
> - We remove "especially in high-dimensional settings" in the revised manuscript.
> - We clarify the definition of the $\tilde{O}$ notation.
> - We carefully check the references and correct all errors/typos you mentioned.
>
> **About *Key Questions For Authors***
>
> **Response to Q1:**  Many thanks for this interesting comment. We fully agree that this is a good direction for further discussion.
>
> Yes, there can be a notion of “best” settings in an appropriate sense. In numerical ODEs, it is common to determine optimal RK by minimizing the leading local truncation error constant. A similar idea can be applied here.
>
> Let us look at the simplest Ornstein–Uhlenbeck (OU) process, given by
> $$dX_{t}= -X_t dt+\sqrt{2} dW_t\,t \ge 0. $$
> The corresponding one-step representation reads
> $$X(t,x;t+h)=x-hx+\sqrt{2}\Delta W_{t,t+h}+\int_{t}^{t+h}\int_{t}^{s}X_rdrds-\sqrt{2}\Delta Z_{t,t+h}.(1) $$
> In this setting, the drift is linear, which allows us to explicitly compute and compare local weak and strong errors.
> The corresponding one-step RKLMC-3G scheme can then be written as
> $$Y(t,x;t+h)=x-hx+\tfrac{h^2}{2}x-\beta a_{11}a_{22}h^3 x+\sqrt{2} \Delta W_{t,t+h}-\sqrt{2}\Delta Z_{t,t+h}+\sqrt{2}\beta a_{22}b_1h\Delta Z_{t,t+h}. (2)$$
> Then
> $$X(t,x;t+h)-Y(t,x;t+h) =\int_{t}^{t+h}\int_{t}^{s}(X_r-x)drds+\beta a_{11}a_{22}h^3x-\sqrt{2}\beta a_{22}b_1h\Delta Z_{t,t+h} =:I_1+I_2+I_3$$
> The error term $I_1$, satisfying
> $$|E[I_1]|\le C_1(d+|x|^2)^{1/2}h^3,(E [ | I_1|^2 ])^{1/2}\le C_2(d+|x|^2)^{1/2}h^{5/2}$$
> is entirely determined by the Langevin  processes, independent of numerical schemes. So we can forget it and just look at the terms $I_2$ and $I_3$, satisfying
> $$|E[I_2]|=|\beta a_{11}a_{22}||x|h^3,(E [ | I_2|^2 ])^{1/2}=|\beta a_{11}a_{22}| |x|h^3,| E [ I_3]| = 0, (E [ | I_3|^2 ])^{1/2}=\tfrac23|\beta a_{22}b_{1}|dh^2.$$
> According to the fundamental strong convergence theorem (see, e.g., Yang and Wang, ICML2025), the principal global mean-squre error should be of order $O(h^{3/2})$, which comes from the term $I_3$. More precisely, on the condition that the local mean error (order $3$) is order $0.5$ higher than the local mean-square error (order $2$), the global mean-square error has a convergence rate of order $1.5$. Therefore, for the above OU process, determining the optimal RKLMC-3G scheme in the sense of minimizing the principal error reduces to the following constrained optimization problem: find method parameters that minimize
> $|\beta a_{22} b_1|$, subject to the order conditions (12)–(14) in this paper. In this sense, the SRK with two gradients with $a_{11} = b_1 = 0$ seems to be the best one, as the above errors all vanishes. But for SRK methods with three gradients, solving the optimization problem is complicated and requires a further investigation.
>
> Please note that the SRK scheme studied in Li et al. (2019) can not be covered by our framework.
>
> As you suggested, we will add some discussions on this in the revised manuscript.
>
> **Response to Q2:** Thanks! Here, saying that a constant is “dimension-independent” means that the constant does not depend on the dimension $d$ of the sampling problem. We will revise the statement of Assumption 2.5 as follows:
>
> Let the target distribution $\pi (\mbox{d} x) \propto e^{-U(x) }\mbox{d} x$
> satisfy the Log-Sobolev inequality with constant $\rho$:
> $$\pi(\phi^2 \log \phi^2)\le\rho \,\pi(|\nabla \phi|^2),\forall \phi \in C_b^1(\mathbb{R}^d), \pi(\phi^2)=1.$$
>
> Here the constant $\rho$ dose not depend on $d$ and such LSI assumption is widely used in the literature. Theorem 2.6 will be revised accordingly as well. Moreover, we will carefully go through the entire manuscript and revise the relevant statements.
>
> **Response to Q3:**  Thanks for your question. We think the observation is indeed an artefact of the experimental settings. By using smaller stepsizes and more Monte Carlo paths to approximate the mean-square errors, we can get rid of this phenomenon. Please note that the error bounds hold as the stepsize $h$ tends to zero.

---

> > ### Author Rebuttal · Reviewer_kJTY · 2026-04-02
> >
> > The authors have addressed my concerns in detail and I thank them for their response. I am maintaining my score of "accept".

---

### Official Review · Reviewer_X85K · 2026-03-12

**Soundness:** 3
**Presentation:** 2
**Significance:** 2
**Originality:** 2
**Overall Recommendation:** 3
**Confidence:** 4

**Summary:**

This paper develops Stochastic Runge-Kutta methods, a class of methods for discretising SDEs, which applied in this setting to Langevin Monte Carlo to sample from a given target density $\pi$. The authors establish a convergence rate with an asymptotic rate $\mathcal{O}(d^{3/2} h^{3/2})$ under log-Sobolev inequality and provide some short numerical section to demonstrate the utility of this method.

**Compliance With Llm Reviewing Policy:**

Affirmed.

**Final Justification:**

Authors did implement some of the things and I believe that in these examples, the algorithm shows some promise. But Bayesian Logistic Regression examples were only conducted on synthetic data. Also the GMM example is a bit simple as the modes are close.

All experiments are being synthetic, there's no realistic ML experiment in this paper. I suggested some real-data benchmarks in a paper referred in my comment (on top, the cited paper include other examples like Funnel distribution etc which is used to test new samplers). To evaluate this (new) sampler, I think some good benchmarking is still necessary. At the moment, a practical user would have no clear reason to choose this algorithm for use (besides its theoretical properties, which is also satisfied by the earlier work SRK-LD with the same order of convergence).

As a result, I'll keep my score.

**Key Questions For Authors:**

1) can authors clarify the nature of their assumptions? In particular, a natural set of assumptions would be (i) dissipativity, (ii) gradient Lipschitzness, and perhaps (iii) Hessian Lipschitzness. On top of this, the authors assume LSI (which could replace dissipativity itself) -- it is not clear to me why both are required. A further assumption on third derivatives also makes things slightly harder to imagine. Please provide a clean explanation of your setting of assumptions.

2) Authors remark:

> We would like to emphasize that all constants used here $(\mu, \mu', L_1, L_1', L_2, L_3)$ are of constant order, independent of the problem dimension $d$.

Would this be true in practice? That is, for a given practical sampling problem, it might be that these constants have hidden dependence to $d$. Please discuss. (see my next question and you can reply both as one).

3) Please provide, a reasonable, practically relevant (not just toy) example that shows all assumptions are satisfied. This has to be added as a remark after assumptions are introduced. This could be posterior distribution in a Bayesian model (e.g. Bayesian logistic regression) to be checked. There has to be one non-log-concave, sensible example where we can clearly see all this smoothness/LSI are verified as well as dissipativity.

The main weakness of this paper in my opinion is the lack of proper numerics to show the impact of RKLMC, and I am afraid this is not quite possible to avoid when introducing a novel scheme with theoretical guarantees. I have some suggestions about numerics below, which I hope authors will find useful:

4) The experimental setting is not clearly written (this is despite authors having space, almost 1.5 pages short of 8 pages). I suggest authors to clarify their first example, which is only a toy example. This is a mixture of Gaussians, multi-modal target (but only two mixtures). What's being evaluated here (what is precisely "mean-square errors"?). The error metrics are not defined properly. Also in Figure 1, it does seem like pink line (which is authors' introduced method) falls short of the required 1.5 rate, whereas prior work SRK-LD doesn't suffer from this issue. Please discuss. Please provide histograms to show that your algorithm is able to sample from both modes reliably (or confirm that this is the case in writing if you cannot share images here).

5) A further comment in the example above, that it mentions that 1st to 3rd order Lip conditions have been verified as well as the log-Sobolev -- please confirm the example satisfies dissipativity, and provide a lemma about it.

6) The authors mention "two numerical experiments" but in fact these are run on the same model. I strongly encourage authors to provide a thorough comparison with the experimental setting in Li et al (2019) where the authors run experiments on Bayesian Logistic Regression + specific non-convex potentials. Please for each example, provide the scaling for $h$ (as you did for the mixture example), scaling for $d$ as you can adjust the dimension.

7) For the bayesian regression example, good benchmarks include Sonar (d = 34) and Ionosphere (d = 61) datasets. Please follow the same methodology as (e.g.) in [1] and follow up works -- and compare your predictive posterior log-likelihoods with the numbers in similar experiments.

[1] Stochastic Localization via Iterative Posterior Sampling

Minor comment: Authors didn't pay attention to the citation style \citet vs \citep. They always seem to use citep: (Li et al., 2019). You need to fix throughout the text, and use citet appropriately only when needed (not always). For example, take the following sentence:

> We mention that our uniform-in-time error analysis differs from that in (Li et al., 2019), where the contractivity of the mean-square error propagation [...]

should indeed be written with citet as

> We mention that our uniform-in-time error analysis differs from that in Li et al. (2019), where the contractivity of the mean-square error propagation [...]

so please fix all of such occurrences.

**Limitations:**

Yes

**Strengths And Weaknesses:**

Strengths:
- Introduction of a new discretisation scheme. This is a useful and novel development, which can have impact just beyond sampling problems, e.g., for diffusion models or interacting particle algorithms
- Very precise theoretical analysis: This is obviously done very properly and more rigorously than the standard ML literature, with moment bounds, precise conditions, explicit constants. It is also clear that the authors tried to keep the assumptions clean.

Weaknesses:
- The intuition about assumptions are not given. The main result assume all 2.1-2.5, which is a lot of assumptions. See my questions about these below.
- Numerical schemes should demonstrate impact in ML for this to have a stronger standing. As the paper does not just provide analysis of an existing (impactful) scheme but it does introduce a new scheme -- and it has to be shown that this brings some critical improvement to some well-known shortcomings of LMC methods.

---

> ### Author Rebuttal · Authors · 2026-03-31
>
> Thank you for the review. Below, we provide a detailed, point-by-point response to each of your concerns.
>
> **About *Weaknesses***
>
> **Response:** Thank you for your comments. Below, we respond to each point in turn.
> - A clearer discussion of the intuition behind the assumptions will be added in the revision; please see our detailed responses below.
> - We will also include additional numerical experiments to further demonstrate the impact of the new scheme in machine learning, specifically the two examples mentioned in Question 6. Please see more details in our Responses to Q6 and Q7.
>
> **About *Key Questions For Authors***
>
> **Response to Q1:** Many thanks for your useful comments. The LSI is used to establish exponential convergence of the dynamics to equilibrium (see Proposition 3.4). Our methodology relies on uniform-in-time moment bounds of the RKLMC algorithms, which can be obtained by crucially using the dissipativity condition (see Propositions 3.1 and 3.2). Using only LSI, we do not have idea to prove the uniform-in-time moment bounds of the RKLMC algorithms. Thus, we need both.
>
> Intuitively, the Lipschitz continuity of the third derivative means that even the variation of the curvature changes in a controlled and non-oscillatory way. We also note that such 1st-to 3rd-order smoothness assumptions are standard in the analysis of higher-order Langevin Monte Carlo methods; see, for example, Li et al. (2019) and Sabanis and Zhang (2019).
>
> **Response to Q2 and Q3:** Thanks. We would first like to emphasize that these assumptions are indeed satisfied in some practically relevant sampling problems. In particular, log-concave distributions satisfy LSI.
>
> As a concrete and practically relevant non-log-concave example, we consider the Bayesian logistic regression (BLR) posterior with potential $U_B$, for which our assumptions can be verified with dimension-independent constants. More precisely, $U_B=U_L+U_P$, where $U_L$ is the negative log-likelihood from the second example in Section 5.1 of Li et al.(2019), and $U_P$ is the Gaussian-mixture negative log-prior used in our numerical experiments.  A detailed verification will be added after the assumptions in the revised manuscript.
>
> **Response to Q4:** Thanks for your helpful suggestions. The mean-square error is the square of  $L^2$-error: $(\mathbb{E}|Y_T^{h}-X_T^{ref}|^2)^{1/2}$. Note that this is a natural metric here since our new RK schemes are analyzed through their mean-square convergence rates and our non-asymptotic $W_2$ bounds also rely on such errors. Due to space limitations, the detailed experimental settings are referred to *Response to Q2* of Reviewer 8nPo.
>
> The pink line in the original figure falls short of order 1.5 because the chosen stepsizes are large. Using smaller stepsizes recovers the expected rate. We will revise the figure accordingly. We also performed a GMM experiment by simulating 5000 SRK-2G trajectories up to T=5. The histogram of the first component shows two clear peaks matching the two modes. This experiment will be added to the revised manuscript.
>
> **Response to Q5:** We apologize for missing this. The considered GMM does satisfy the dissipativity condition with $\mu=1/2$ and $\mu'd=2|\mu_1|^2$. We will add a lemma in the revised manuscript to make this explicit; the detailed proof follows Proposition 3.1 of Neufeld and Zhang (2025, JMAA).
>
> **Response to Q6:** Thanks for this valuable suggestion. Following your comment, we added numerical experiments for the Bayesian logistic regression (BLR) model and the specific non-convex potential (NCP) in Li et al. (2019), comparing LMC, SRK-LD, and SRK-2G in terms of convergence with respect to the stepsize h. The results are consistent with the GMM example: LMC shows order-1 convergence, both SRK-LD and SRK-2G show order-1.5 convergence.
>
> We also examined the dimension dependence of SRK-LD and SRK-2G for the GMM, BLR and NCP. For the GMM and BLR, the observed dependence is approximately $d^{1.5}$, in agreement with the theory, for the NCP we get, around $d^{0.8}$ for SRK-LD and SRK-2G, maybe because it is a quadratic polynomial in $|x|$.
>
> Due to space limitations, we omit the error table here. If you would like to see it, please let us know and we will include them in the next rebuttal. As suggested, we will incorporate these additional experiments and the related discussion into the revised manuscript.
>
> **Response to Q7:** Thanks for bringing this paper to our attention. Following your suggestion, we tested SRK-LD, SRK-2G and LMC on these two datasets using the average predictive posterior log-likelihood. All these three schemes show worse SLIPS values than those reported in the excellent paper. We wil cite it. Based on SRK-2G, we hope to make further progress on this aspect in the future.
>
> **Response to *Minor comment*:** Thanks. Following your comments, we will thoroughly check the entire manuscript and revise it accordingly. If you had any further question, please let us know.

---

> > ### Author Rebuttal · Reviewer_X85K · 2026-04-03
> >
> > Thank you for your reply. If I am not mistaken, In Q4 and Q6, authors promise some plots about experiments, but these plots are not shared. In order to fully resolve my concerns, I suggest authors to share these plots and the experimental setting, in an appropriate form. They can use anonymous links to share images as this is allowed.

---

> > > ### Author Response · Authors · 2026-04-06
> > >
> > > Thank you for your reply. Via the anonymous link: https://anonymous.4open.science/r/RKLMC-60D0/README.md,  we share plots on experimental results mentioned in our previous responses to Q4 and Q6. Below, we briefly describe the experimental settings and summarize the numerical results.
> > >
> > > ## Numerical Experiment Settings
> > > - **Gaussian Mixture Model (GMM).** For this experiment, we consider a two-component Gaussian mixture target,
> > > where the two modes are located at $\mu_1$ and $\mu_2=-\mu_1$ with $|\mu_1|=2$, unit variances and mixture weights $0.5$ and $0.5$.
> > > - **Bayesian Logistic Regression (BLR).** For this experiment, we consider a Bayesian logistic regression posterior based on synthetic data. The prior strength is set to $\alpha=0.5$. For a given dimension $d$, the true parameter is chosen as $\theta_{true}=\frac{1}{\sqrt d}I_d$. The design matrix $X\in\mathbb{R}^{n\times d}$ is generated with i.i.d.standard Gaussian entries with $n=100$. The response variables are then sampled from the logistic model with success probabilities
> > > $p_i=1/(1+\exp(-x_i^\top\theta_{true}))$, that is, $Y_i\sim \mathrm{Bernoulli}(p_i)$. Define the empirical covariance matrix by $\Sigma_X=\frac{X^\top X}{n}$.
> > > - **Non-Convex Potential (NCP).** For this experiment, we consider the target distribution considered in Li et al. (2019), with the non-convex potential $U(x)=\sqrt{\beta+\|x\|^2}+\gamma \log(\beta+\|x\|^2)$, where $\beta=0.3$ and $\gamma=-1$. Then we infer $\nabla^2 U(0)=(\frac{1}{\sqrt{\beta}}+\frac{2\gamma}{\beta})I_d=(\frac{1}{\sqrt{0.3}}-\frac{2}{0.3})I_d<0$. Hence, the potential is non-convex.
> > >
> > > **Probability Histogram** In this experiment, we run the SRK-2G algorithm for the Langevin diffusion to illustrate its sampling performance. The scheme starts from the initial point $X_0=0$ and is simulated up to terminal time $T=5$. We generate $M=5000$ independent trajectories on a fine grid with stepsize $h=2^{-14}$. After evolving the SRK-2G scheme to time $T=5$, we record the first component of the terminal samples. We then plot the probability histogram of these samples and compare it with the exact one-dimensional marginal density of the GMM.
> > >
> > > From the file *RKLMC\_GMM\_PDF.pdf*, we observe that the empirical distribution produced by SRK-2G matches the exact marginal density well.
> > >
> > > **Convergence Rates and Dimension Dependence**
> > > - 1.Common Experimental Settings
> > >
> > > For all three examples, all schemes start from the same initial data $X_0=0$ and are simulated up to terminal time $T=2.$ In each experiment, we generate $M=5000$ independent trajectories. To ensure a fair comparison, we generate a common Brownian path on a very fine grid with stepsize $h_{\mathrm{ref}}$ and construct all coarse approximations from this same path. The reference solution (RS) is taken to be the finest-grid LMC trajectory with stepsize $h_{ref}$.  The error is measured by the terminal root mean-square error $\left(\mathbb{E}|Y_T^h-X_T^{\mathrm{ref}}|^2\right)^{1/2}$, where $Y_T^h$ is the terminal value of the numerical scheme (NS) and $X_T^{\mathrm{ref}}$ is the reference solution. For the convergence rate and dimension dependence experiments, we plot the root mean-square errors against the stepsizes and dimensions, respectively, on a log-log scale.
> > > - 2. Different Experimental Settings
> > >
> > > **Convergence Rate**
> > > | Model | Dimension | Stepsize of RS | Stepsizes of NS |
> > > |---|---|---|---|
> > > | GMM | $d=10$ | $h_{ref}=2^{-15}$ | $h = 2^{-10}, \ldots, 2^{-6}$ |
> > > | BLR | $d=10$ | $h_{ref}=2^{-15}$ | $h = 2^{-10}, \ldots, 2^{-6}$ |
> > > | NCP | $d=10$ | $h_{ref}=2^{-15}$ | $h = 2^{-9}, \ldots, 2^{-5}$ |
> > >
> > > From files *RKLMC\_xxx\_CR.pdf*, we observe that for all three models, namely GMM, BLR and NCP, the convergence rate of LMC is approximately order $1$,  those of SRK-LD and SRK-2G are approximately order $1.5$.
> > >
> > > **Dimension Dependence**
> > >
> > > | Model | Dimensions | Stepsize of RS | Stepsize of NS |
> > > |---|---|---|---|
> > > | GMM | $d=8,10,12,14,16$ | $h_{ref}=2^{-13}$ | $h = 2^{-5}$ |
> > > | BLR | $d=6,8,10,12,14$ | $h_{ref}=2^{-11}$ | $h = 2^{-6}$ |
> > > | NCP | $d=4,6,8,10,12$ | $h_{ref}=2^{-11}$ | $h = 2^{-5}$ |
> > >
> > > From files *RKLMC\_xxx\_DD.pdf*, we observe that for the GMM and BLR models, the dimension dependence of SRK-LD and SRK-2G  is approximately $d^{1.5}$. For the NCP,  it is approximately  $d^{0.8}$ for SRK-LD and SRK-2G.

---

### Decision · Program_Chairs · 2026-04-30

**Decision:**

Accept (regular)

**Comment:**

This paper introduces a Langevin Monte Carlo sampling algorithm based on a stochastic Runge-Kutta scheme, which is computationally cheaper than an existing sampler based on Runge-Kutta, and presents non-asymptotic error bounds on the algorithm along with numerical results. Reviewers overall praised the novelty of the work and appreciated the theoretical contributions of this work

Because this paper proposes a new algorithm (rather than an analysis of an existing algorithm), the main weakness brought forth by Reviewer X85K was the lack of a substantial ML experiment. While the reviewer appreciated the new experiments provided during the rebuttal, they wanted to see experiments at the scale of the benchmarks suggested in the initial review. Despite this, other reviewers felt the strength of the theoretical results and the new experiments provided during the rebuttal made the paper a notable contribution.

Please use the remaining pages allowed to expand upon the empirical results, including clarifying the empirical setting as stated by Reviewer X85K and to clarify the assumptions of the theory. Finally, reviewers have pointed out a number of typos and notation errors that should be corrected before publication.